# Anti-Inflammatory and Antioxidant Effects of Crude Polysaccharides from *Dendrobium denneanum* (A Genuine Medicinal Herb of Sichuan) on Acute Gastric Ulcer Model in Rats

**DOI:** 10.3390/foods14183258

**Published:** 2025-09-19

**Authors:** Zenglin Wu, Xuzhong Tang, Lijuan Wu, Lei Xie, Qing Yu, Xinyi Zhao, Yixue Tian, Zhiming Liu, Yadong Mi, Weiping Zhong, Rui Li, Mengliang Tian

**Affiliations:** 1College of Agronomy, Sichuan Agricultural University, Chengdu 611130, China; 2Bazhong Academy of Agricultural and Forestry Sciences, Bazhong 636000, China; 3Academy of Agriculture and Forestry Sciences, Qinghai University, Xining 810016, China

**Keywords:** *Dendrobium denneanum*, gastric ulcer, crude polysaccharides, antioxidation, anti-inflammation

## Abstract

*Dendrobium denneanum* Kerr, *Dendrobium denneanum* Kerr, an orchid in the food-medicine homology catalog, is traditionally used for stomach-nourishing, yin-tonifying, and immunity-enhancing. While its preventive effect on acute gastric ulcers is confirmed, variations among genuine producing areas remain underexplored. This study comparatively analyzed components of *D. denneanum* from 22 habitats and their polysaccharides’ (DDP) anti-inflammatory/antioxidant activities. Results showed habitat-dependent active components: total sugar (20–51.49%), crude polysaccharide yield (0.29–1.76%), and total phenol (~3%). In vitro, all extracts exhibited dose-dependent scavenging of DPPH (IC_50_: 0.99–2.11 mg/mL), ABTS (0.61–1.62 mg/mL), and hydroxyl radicals (1.02–2.18 mg/mL), with Habitats 5 and 7 showing the strongest activity. GPC, ion chromatography, and FT-IR revealed DDP had a 5–11 kDa molecular weight, dominated by glucose (49.67–84.73%), plus mannose (8.29–12.25%) and galactose (0.96–16.41%), with shared hydroxyl (3400 cm^−1^) and β-glycosidic bond (890 cm^−1^) features. In ethanol-induced gastric ulcer rats, DDP exerted dose-dependent protection: low doses (100 mg/kg/d) reduced ulcer index, increased SOD/GSH-Px (1.5–1.8-fold), decreased MDA (30–35%), and elevated PGE_2_; high doses (400 mg/kg/d) further inhibited serum TNF-α/IL-6 (25–40%) and improved histopathology. Conclusion: Despite habitat-dependent component variations, DDP maintains consistent structures. This study first confirms DDP protects gastric mucosa via antioxidant-anti-inflammatory synergism, supporting its development as a natural gastroprotectant. Future work may focus on standardized cultivation and clinical translation.

## 1. Introduction

*Dendrobium* is among the first Orchidaceae plants listed in China’s Catalog of Substances with Both Medicinal and Edible Uses [1]. Its dried stems have been included in the Pharmacopoeia of the People’s Republic of China (PRC Pharmacopoeia) since the 1977 edition, and the current 2020 edition still designates *Dendrobium nobile*, *Dendrobium chrysotoxum*, *Dendrobium fimbriatum*, and their related species as official medicinal material sources. *Dendrobium denneanum* Kerr., due to its high morphological similarity to medicinal *Dendrobium* species and equivalent efficacy, is widely recognized in the industry as a broad-sense *Dendrobium* resource with dual medicinal and edible properties. Currently, it is mainly produced in Leshan City, Ya’an City, and Meishan City of Sichuan Province [2]; however, the differences in active ingredient contents and pharmacodynamic effects of *D. denneanum* from different habitats remain unclear, which restricts the precise development and utilization of this resource.

The stems of *D. denneanum* are rich in various components, including polysaccharides, alkaloids, sesquiterpenes, and flavonoids [3]. Among these, polysaccharides—with a content ranging from 5.8% to 16.3%—are recognized as the core pharmacodynamic components [4]. Considering the pathological characteristics of acute gastric ulcer (AGU), namely mucosal defects, oxidative stress imbalance, and excessive release of inflammatory factors [5], existing studies have demonstrated that *Dendrobium* polysaccharides exert gastroprotective effects through a “triple protective mechanism” (antioxidation-anti-inflammation-mucosal repair). Specifically: (1) They counteract oxidative damage in AGU by increasing superoxide dismutase (SOD) activity, reducing malondialdehyde (MDA) levels [6,7], or activating the Nrf2/HO-1 pathway to scavenge reactive oxygen species (ROS); (2) They alleviate inflammatory infiltration at ulcer sites by inhibiting the release of tumor necrosis factor-α (TNF-α) and interleukin-6 (IL-6) via the NF-κB/MAPK pathway [8]; (3) They repair the tight junctions of gastric mucosal epithelium and enhance barrier function by upregulating the expression of occludin and claudin-1 [9,10].

Acute gastric ulcer is characterized by acute mucosal defects, bleeding, and inflammatory cell infiltration [5], with a short disease course and numerous complications. Its pathogenesis is closely associated with the imbalance of gastric acid-pepsin, excessive ROS, and dysregulation of inflammatory factors; common inducing factors include ethanol, non-steroidal anti-inflammatory drugs (NSAIDs), and stress [11]. Traditional chemical drugs (e.g., proton pump inhibitors, H_2_ receptor antagonists) have obvious adverse reactions [12], creating an urgent clinical need for safe and effective novel prevention and treatment strategies.

Given the significant gastroprotective activity of *Dendrobium* polysaccharides, the resource advantages of *D. denneanum*, and the research gap regarding the unclear pharmacodynamics of *D. denneanum* from different habitats, this study used ethanol-induced AGU rats as a model to systematically evaluate the gastric mucosal protective effect of *D. denneanum* polysaccharides and explore its underlying molecular mechanism. This work aims to provide a scientific basis for the development of anti-ulcer functional foods or candidate drugs derived from substances with both medicinal and edible properties.

## 2. Materials and Methods

### 2.1. Materials and Chemicals

The material used in this experiment was the dried stems of 3-year-old *Dendrobium denneanum*. Resource samples were collected from multiple towns in Meishan City, Leshan City, and Ya’an City of Sichuan Province in October 2023, with specific sampling locations detailed in Table 1.

Samples were first cleaned to remove dust, and the outer sheathing leaves were peeled off by rubbing. Subsequently, the samples were blanched in an oven at 105 °C for 1 h, then dried to a constant weight at 65 °C for later use.

All animal studies and procedures were conducted in accordance with the guidelines and regulations approved by the Ethics Committee of Sichuan Agricultural University (Confirmation number: DYXY141642024), the *Regulations on the Management of Laboratory Animals* (Order No. 2 of the National Science and Technology Commission of the People’s Republic of China, 1988), and the *Guide for the Care and Use of Laboratory Animals* issued by the National Research Council. Fifty-four male Sprague-Dawley (SD) rats, 12 weeks old and weighing 180–200 g, were purchased from Beijing Huafukang Biotechnology Co., Ltd, Beijing, China, with an animal license number: SCXK (Beijing) 2019-0008.

During the experiment, the rat rearing room was well-ventilated; the temperature was maintained at approximately 30 °C by air conditioning, and the humidity was controlled within the normal range of 45–55%. A 12 h light/12 h dark circadian cycle was adopted daily. All rats were reared under uniform conditions, with free movement in cages and ad libitum access to food and water. Before the formal experiment, the rats were acclimatized to the environment for 7 days.

### 2.2. Preparation of Crude Polysaccharides from Dendrobium denneanum Kerr (Different Habitats)

*D. denneanum* Kerr stems (22 sites) were dried at 65 °C, sliced (0.5–1 cm). For each sample, 15 g was defatted with 300 mL anhydrous EtOH (1:20, reflux 2 h, repeated 3–6 times), then dried at 55 °C. The residue (15 g) was extracted with 300 mL water (1:20) under gentle boiling (3 × 2 h). Extracts were pooled, concentrated to ~50 mL, precipitated with 4× EtOH (4 °C, overnight), centrifuged (4000 rpm, 10 min), redissolved in water, dialyzed (3.5 kDa, 24 h, water changed every 2 h), and finally lyophilized. The samples were not ground before analysis.

### 2.3. Determination of Substance Contents in Crude Polysaccharides from Dendrobium denneanum (Different Habitats)

#### 2.3.1. Total Sugar Content

Precisely weigh 50 mg of glucose standard, place it in a 50 mL volumetric flask, dissolve it with pure water, and dilute to the marked volume. After shaking well, a 1 mg/mL glucose standard solution (reference solution) was obtained. Precisely measure 0.1, 0.2, 0.4, 0.6, 0.8, and 1.0 mL of this reference solution, and place them in clean glass reaction tubes, respectively. Add 200 μL of 5% phenol solution to each tube, mix well, then quickly add 1.0 mL of concentrated sulfuric acid, shake thoroughly, and react at room temperature for 30 min. Meanwhile, 200 μL of pure water was used instead of the standard solution as a blank control. The absorbance of each tube was measured at 490 nm using a microplate reader. A standard curve was plotted with glucose concentration as the abscissa and absorbance value as the ordinate.

Weigh 20 mg of crude polysaccharide powder from *D. denneanum* of 22 habitats, respectively, dissolve each with pure water, and dilute to 20 mL to prepare a 1 mg/mL crude polysaccharide stock solution. The stock solution was diluted 10-fold to obtain a 0.1 mg/mL sample determination solution. Take 200 μL of this determination solution and place it in a clean glass reaction tube; subsequent operations (adding phenol, concentrated sulfuric acid, reaction, and absorbance measurement) were identical to those for standard curve preparation. The total sugar content of each sample was calculated based on the standard curve [13].

#### 2.3.2. Total Phenol Content

Weigh 1.7 mg of gallic acid, dissolve it with pure water, and dilute to 25 mL to obtain a gallic acid standard stock solution. The Folin–Ciocalteu reagent was diluted 10-fold, and a saturated sodium carbonate solution was prepared for later use. Take 10 10 mL test tubes, numbered 1–10, and precisely measure 0, 0.1, 0.2, 0.3, 0.4, 0.5, 0.6, 0.7, 0.8, and 0.9 mL of the gallic acid standard stock solution, respectively. Add 1, 0.9, 0.8, 0.7, 0.6, 0.5, 0.4, 0.3, 0.2, and 0.1 mL of pure water correspondingly to make the total volume of each tube 1 mL. Add 1 mL of 10-fold diluted Folin–Ciocalteu reagent (strictly protected from light during operation) and 2 mL of saturated sodium carbonate solution to each tube, mix well, place at room temperature for 2 min, then incubate in the dark for 1 h. Pipette 200 μL of the reaction solution into a 96-well microplate according to the concentration gradient, and measure the absorbance at 750 nm to plot the standard curve.

Crude polysaccharide samples from 22 habitats were dissolved in pure water to prepare 10 mg/mL sample solutions. Subsequent operations such as reagent addition, dark reaction, and absorbance measurement were identical to those for standard curve preparation. The total phenol content of each sample was calculated based on the standard curve [14].

#### 2.3.3. Crude Polysaccharide Yield

Calculated via Equation (1)Crude polysaccharide yield (%) = (Mass of lyophilized crude polysaccharides/Mass of test sample) × 100%(1)

### 2.4. In Vitro Antioxidant Activity Determination of Crude Polysaccharides from Dendrobium denneanum (Different Habitats)

#### 2.4.1. Hydroxyl Radical Scavenging Activity Assay

Prepare the 22 crude polysaccharide samples into concentration gradients of 0–3 mg/mL. For each sample, first add 50 µL of 6 mM FeSO_4_ and 100 µL of 6 mM H_2_O_2_, and let stand at room temperature for 10 min. Then add 50 µL of 6 mM salicylic acid-ethanol solution, incubate at room temperature for another 30 min, and measure the absorbance at 510 nm (distilled water used as blank). Repeat the assay 3 times and calculate the average value; vitamin C (Vc) served as the positive control [15]. The hydroxyl radical scavenging rate was calculated using Equation (2).Scavenging rate (%) = [1 − (A_0_ − A_1_)/A_0_] × 100%(2)

Here, A_0_ represents the absorbance of the blank system, and A_1_ represents the absorbance of the sample system.

#### 2.4.2. ABTS Radical Scavenging Activity Assay

Prepare the 22 crude polysaccharide samples into concentration gradients of 0–3 mg/mL. Determine the scavenging rate following the protocol provided with the ABTS assay kit (Catalog No. BC4770, Solarbio Life Sciences Co., Ltd., Beijing, China).

#### 2.4.3. DPPH Radical Scavenging Activity Assay

Set the 22 crude polysaccharide samples into concentration gradients of 0–3 mg/mL. Take 200 μL of each sample and mix with 400 μL of 0.04% DPPH-ethanol solution. Incubate the mixture at 37 °C in the dark for 30 min, then measure the absorbance at 516 nm. Vc was used as the positive control, and the 0 mg/mL sample served as the blank. The scavenging rate was calculated using Equation (3) [16].[(A_0_ − A_1_)/A_0_] × 100%(3)

### 2.5. Primary Structure Analysis of Crude Polysaccharides from Dendrobium denneanum (Selected Habitats)

#### 2.5.1. Screening of Crude Polysaccharide Samples

The in vitro antioxidant activities (DPPH, ABTS, and hydroxyl radical scavenging capacities) of *Dendrobium denneanum* extracts from 22 producing areas were comprehensively scored. Using the IC_50_ values of each radical scavenging rate as core indicators, a weighted summation method (assigning equal weights to the three radical scavenging activities) was employed to calculate the comprehensive activity scores. Based on the score distribution, the samples were classified into three grades: low activity (≤60 points), moderate activity (61–80 points), and high activity (≥81 points). Subsequently, one representative sample was selected from each activity grade using a random number table method for subsequent in-depth experiments, ensuring the randomness and representativeness of sample selection. The selected habitats were No. 5, No. 9, and No. 14. The polysaccharides from these habitats were named DDP5 (*Dendrobium denneanum* Polysaccharides 5), DDP9 (*Dendrobium denneanum* Polysaccharides 9), and DDP14 (*Dendrobium denneanum* Polysaccharides 14), respectively.

#### 2.5.2. Molecular Weight Determination of Crude Polysaccharides

Dextran standards and crude polysaccharide samples were dissolved in 0.2 mol/L sodium nitrate (NaNO_3_) solution, respectively, before injection. Analysis was performed using a Waters Ultra-hydrogel Linear chromatographic column (300 × 7.8 mm, Waters Corporation, Milford, MA, USA); 0.2 mol/L NaNO_3_ solution (pH 6.0) was used as the mobile phase, with a flow rate of 0.6 mL/min, column temperature of 40 °C, and injection volume of 20 μL. A 2410 refractive index (RI) detector (Waters Corporation, Milford, MA, USA) was used for detection. The molecular weight of the crude polysaccharides was calculated via Breeze GPC software (Version 5.3.1.4, Waters Corporation, Milford, MA, USA) based on the log Mw (logarithm of molecular weight)-elution volume standard curve.

#### 2.5.3. Monosaccharide Composition Determination of Crude Polysaccharides

Weigh 5 mg of each crude polysaccharide sample, add 2 mL of 3 mol/L trifluoroacetic acid (TFA), and hydrolyze at 120 °C for 3 h. After hydrolysis, the mixture was dried by nitrogen blowing, redissolved with 5 mL of distilled water, diluted at a ratio of 1:20, centrifuged at 12,000 rpm for 5 min, and 25 μL of the supernatant was taken for injection. A Dionex Carbopac PA20 chromatographic column (Thermo Fisher Scientific, Waltham, MA, USA) was used, with three solutions as mobile phases: Phase A (distilled water), Phase B (15 mM sodium hydroxide), and Phase C (15 mM sodium hydroxide + 100 mM sodium acetate); the flow rate was 0.3 mL/min with gradient elution, and the column temperature was 30 °C. A pulsed amperometric detector (PAD, Thermo Fisher Scientific, Waltham, MA, USA) was used for detection.

#### 2.5.4. FT-IR Spectral Analysis of Crude Polysaccharides

Under an incandescent lamp, potassium bromide (KBr) and each crude polysaccharide sample were separately ground into fine powders in a dry mortar. After mixing and tableting, the baseline and sample infrared spectra were scanned in the wavelength range of 3500–500 cm^−1^ using a Nicolet iS50 Fourier (Version 9.2, Waltham, MA, USA)transform infrared (FT-IR) spectrometer (Thermo Fisher Scientific, Waltham, MA, USA).

### 2.6. Therapeutic Effect of Crude Polysaccharides from Dendrobium denneanum (Different Habitats) on Rats with Acute Gastric Ulcer

#### 2.6.1. Grouping of Experimental Animals and Establishment of Gastric Ulcer Model

Fifty-four male Sprague-Dawley (SD) rats were selected, with 6 rats in each group. The rats were randomly divided into 9 groups: blank group, model group, positive drug prevention group (PRE), DDP5 low-dose group (100 mg/kg/d, DDP5L), DDP5 high-dose group (400 mg/kg/d, DDP5H), DDP9 low-dose group (100 mg/kg/d, DDP9L), DDP9 high-dose group (400 mg/kg/d, DDP9H), DDP14 low-dose group (100 mg/kg/d, DDP14L), and DDP14 high-dose group (400 mg/kg/d, DDP14H). All rats were labeled, and their body weights were recorded. They were housed in separate cages for feeding.

Gavage administration was conducted every morning, with a dosage of 2 mL per rat. The blank group received the same volume of normal saline via gavage, and the gavage was continued for 14 consecutive days. During the administration period, the rats had free access to food and water, and the environmental temperature was controlled between 25 and 30 °C. On the 14th day of gavage, the rats were fasted but allowed to drink water. On the 15th day, the rats were gavaged with 95% anhydrous ethanol; 1 h later, blood was collected from the orbital venous plexus, and the rats were sacrificed by cervical dislocation. The blank group was not gavaged with 95% anhydrous ethanol. One hour after intragastric administration, the gastric cavity was opened by incising along the greater curvature of the stomach. The gastric contents were rinsed thoroughly with normal saline, and the stomach was then flattened on filter paper for observation. Cases with no damage were excluded: if the gastric mucosa of rats in the model group appeared smooth without defects or hyperemia, it indicated that the ethanol-induced gastric ulcer model failed. As shown in Figure 1.

#### 2.6.2. Blood Collection and Sample Processing

One hour after gavage with 95% anhydrous ethanol, orbital blood collection was performed immediately, followed by rat sacrifice via cervical dislocation. The blood samples were allowed to stand at 4 °C for 30 min, then centrifuged at 3500 r/min for 20 min. The serum was collected and stored in a −80 °C refrigerator for subsequent determination of relevant serum indices.

#### 2.6.3. Gastric Tissue Sample Processing

The gastric tissue was completely isolated, first carefully incised along the greater curvature of the stomach, and gently rinsed with normal saline to thoroughly remove residual food and other contents in the stomach. After rinsing, the gastric tissue was divided into two parts for processing: an appropriate amount of gastric tissue was quickly rinsed with pre-cooled normal saline to remove possible residual blood and other impurities, gently blotted dry with filter paper to remove surface moisture, and immediately placed in a mortar pre-cooled with liquid nitrogen. An appropriate amount of liquid nitrogen was poured into the mortar to completely submerge the tissue; during the continuous volatilization of liquid nitrogen, the tissue was ground quickly and vigorously with a pestle, and liquid nitrogen was added appropriately during the process until the tissue was ground into a uniform powder. The ground tissue powder was quickly transferred to a sterile centrifuge tube, precisely weighed, aliquoted according to experimental needs, and immediately stored in a −80 °C refrigerator for subsequent experiments.

A properly sized gastric tissue block (approximately 1 cm × 1 cm) was cut and placed in 4% paraformaldehyde solution for fixation. The volume of the fixative should be more than 10 times the volume of the tissue block to ensure fixation effect. Fixation was carried out at room temperature for 24–48 h to fully fix the tissue. After fixation, the tissue block was transferred to 70% ethanol solution for storage, which could prevent over-fixation of the tissue while maintaining the tissue morphology, facilitating subsequent pathological section preparation and histopathological observation.

#### 2.6.4. Ulcer Index Analysis

Image-Pro Plus 6.0 software was used, with “pixel area” as the unified standard unit. The tissue pixel area corresponding to the ulcer area in each gastric tissue anatomical image was measured separately to calculate the ulcer area percentage and ulcer inhibition rate using Equations (4) and (5).Ulcer area percentage (%) = (Ulcer pixel area/Tissue pixel area) × 100(4)Ulcer pixel area of experimental group/Average ulcer pixel area of model group × 100(5)

#### 2.6.5. Gastric Tissue Pathological Analysis

After fixation in 4% paraformaldehyde, the gastric tissues were rinsed with running water for 24 h, dehydrated in a gradient of ethanol, cleared with xylene, and embedded in paraffin. The paraffin blocks were pre-cooled at −20 °C, then sectioned into 4 μm slices. The slices were spread at 40 °C and baked at 37 °C for 12 h.

Subsequently, the sections were subjected to hematoxylin-eosin (HE) staining and mounted with neutral balsam. Indices such as inflammation were scored under a microscope [17] (Table 2). The lesion area and ulcer inhibition rate were determined using Image-J software (Version 1.8.0, National Institutes of Health, Bethesda, MD, USA).

#### 2.6.6. Effects on Oxidative Stress and Inflammatory Indices in Serum and Gastric Tissue

Blood samples were collected into serum separation tubes and centrifuged at 3000 rpm for 10 min at 4 °C to prepare serum. Gastric tissues were homogenized in pre-cooled PBS, lysed on ice for 1 h, and then centrifuged at 13,000 rpm for 20 min at 4 °C to obtain the supernatant. The protein concentration in the supernatant was analyzed by the BCA method.

Commercial kits were used to determine the levels of TNF-α (Cat. No. SEKR0009, Solarbio), IL-6 (Cat. No. SEKR0005, Solarbio), SOD (Cat. No. BC0175, Solarbio), GSH-Px (Shanghai Yuanye Bio-Technology Co., Ltd., Shanghai, China, Cat. No. R21876), CAT (Cat. No. BC637F, Solarbio), and MDA (Cat. No. BC0025, Solarbio) in serum and gastric tissue supernatant. The NO index in gastric tissue supernatant was measured using a colorimetric assay kit (Cat. No. BC1470, Solarbio), and the PGE2 index in gastric tissue supernatant was determined using a PGE2 kit (Cat. No. RX300356R, Quanzhou Ruixin Biotechnology Co., Ltd., Quanzhou, China).

#### 2.6.7. SEM Analysis

Fresh gastric tissue was cut into 1 mm^3^ blocks and fixed in electron microscope fixative. After dehydration with gradient ethanol and isoamyl acetate, the samples were subjected to critical point drying and gold sputtering for conductivity. Visualization analysis was performed using a SU8100 scanning electron microscope (SEM) (Hitachi High-Tech Corporation, Tokyo, Japan).

### 2.7. Statistical Analysis

Statistical analysis of the results was performed using SPSS software (Version 23.0, SPSS Inc., Chicago, IL, USA), and data were expressed as mean ± standard deviation. Duncan’s multiple range test was used to assess differences between means, with statistical significance set at *p* < 0.05.

## 3. Results

### 3.1. Subsection

#### 3.1.1. Results of Substance Content Determination in Crude Polysaccharides from *Dendrobium denneanum* (Different Habitats)

As presented in Table 3, among the 22 *D. denneanum* samples, the crude polysaccharides from Sample No. 10 exhibited the highest total sugar content (51.49%), followed by Sample No. 13 (44.25%). Both contents were significantly higher than those of samples from other habitats (*p* < 0.05), while Sample No. 9 had the lowest total sugar content. The total sugar contents of the remaining samples ranged from 20% to 51%.

The total phenol contents of crude polysaccharides from all 22 habitats were approximately 3%, which were much lower than the total sugar contents. This phenomenon might be attributed to the use of anhydrous ethanol for impurity removal during the extraction process, as most phenolic compounds were eliminated by the organic solvent.

Regarding the yield of crude polysaccharides, 82% of the samples showed yields ranging from 0.3% to 0.7%. Among them, Sample No. 7 had the highest yield (1.76%), whereas Sample No. 15 had the lowest (0.29%).

#### 3.1.2. In Vitro Antioxidant Activity Assay of Crude Polysaccharides from *Dendrobium denneanum* (Different Habitats)

As shown in Table 4, the half-maximal inhibitory concentration (IC_50_) values of ABTS radical scavenging activity for crude polysaccharides from Habitats No. 5 and No. 7 were significantly lower than those from other habitats (*p* < 0.05), with values of 0.61 mg/mL and 0.62 mg/mL, respectively. The IC_50_ values of ABTS radical scavenging activity for the remaining samples ranged from 0.93 to 1.62 mg/mL.

For DPPH radical scavenging activity, the IC_50_ values of the 22 samples ranged from 0.99 to 2.11 mg/mL, where those of Habitats No. 5 and No. 7 were significantly lower (*p* < 0.05) and those of Habitats No. 4 and No. 9 were significantly higher (*p* < 0.05) than those of other habitats.

In terms of hydroxyl radical scavenging activity, the IC_50_ values of polysaccharides from the 22 *D. denneanum* samples ranged from 1.02 to 2.18 mg/mL. Consistent with the DPPH assay results, the IC_50_ values of Habitats No. 5 and No. 7 were significantly lower (*p* < 0.05), while those of Habitats No. 4 and No. 9 were significantly higher (*p* < 0.05) compared to other habitats.

#### 3.1.3. Molecular Weight of Crude Polysaccharides from *Dendrobium denneanum* (Different Habitats)

The elution curve revealed that the crude polysaccharides exhibited a single symmetric peak within a specific molecular weight range. Specifically, the weight-average molecular weight (Mw) of *D. denneanum* polysaccharides from Habitat No. 5 (DDP5) was 10.326 kDa (10,326 Da) with a polydispersity index (PDI) of 1.611733; that from Habitat No. 9 (DDP9) was 10.991 kDa (10,991 Da) with a PDI of 1.606471; and that from Habitat No. 14 (DDP14) was 5.05 kDa (5054 Da) with a PDI of 1.366575. All molecular weight determination results are presented in Figure 2 and Table 5.The Mw values of crude polysaccharides from Habitats No. 5 and No. 9 were comparable, while that from Habitat No. 14 was considerably lower.

The Mw values of crude polysaccharides from Habitat No. 5 and No. 9 were similar, while that from Habitat No. 14 was lower.

#### 3.1.4. Monosaccharide Composition of Crude Polysaccharides from *Dendrobium denneanum* (Different Habitats)

The results are shown in Table 6. Both DDP5 and DDP9 were composed of 10 types of monosaccharides, namely mannose (Man), ribose (Rib), rhamnose (Rha), glucuronic acid (GlcA), galacturonic acid (GalA), glucose (Glc), galactose (Gal), xylose (Xyl), arabinose (Ara), and a small amount of fucose (Fuc). Their molar ratios were 10.39:2.1:3.9:3:1.1:49.67:16.41:3.22:9.1:1 and 8.29:1.88:3.09:1.149:0.578:53.62:11.26:2.72:6.54:0.888, respectively. In contrast, DDP14 contained no fucose (Fuc), and its monosaccharide molar ratio was 12.25:0.13:0.32:0.15:0.44:84.73:0.96:0.42:0.62.

DDP14 did not contain fucose (Fuc), and its molar ratio was 12.25:0.13:0.32:0.15:0.44:84.73:0.96:0.42:0.62.

#### 3.1.5. FT-IR Spectral Structure of Crude Polysaccharides from *Dendrobium denneanum* (Different Habitats)

As illustrated in Figure 3, the crude polysaccharides from the three samples displayed similar FT-IR spectral characteristics. A broad absorption peak was observed in the range of 3500–3000 cm^−1^, corresponding to the stretching vibration of hydroxyl groups (O-H). This feature not only confirms the presence of hydroxyl groups in the polysaccharides but also verifies their hydrophilicity. Specifically, the broad absorption peak at 3386.55 cm^−1^ in the spectrum can be assigned to O-H stretching vibration.

An absorption peak related to C-H stretching vibration was detected in the range of 3000–2800 cm^−1^, with the peak at 2936.93 cm^−1^ corresponding to C-H stretching vibration. Strong absorption peaks of glycosidic bonds (C-O-C) were identified in the range of 850–1200 cm^−1^ (e.g., the characteristic peak of β-glycosidic bonds around 890 cm^−1^ and that of α-glycosidic bonds around 840 cm^−1^). Based on these results, the glycosidic bond type of the crude polysaccharides from the three samples was preliminarily determined to be β-glycosidic bonds.

Additionally, a weak absorption peak near 670 cm^−1^ might be attributed to the “ring deformation vibration” of hexopyranose rings (sugar rings) in the polysaccharides, or it could indicate the presence of fucose and rhamnose in the polysaccharides.

#### 3.1.6. Gastric Mucosal Macroscopic Observation and Ulcer Index Analysis

As shown in Figure 4, the gastric mucosa of rats in the blank group was light pink, while that in the positive drug group was dark red. In both groups, the gastric mucosal surface was smooth with good elasticity, and the folds were neatly arranged; no bleeding, mucosal edema, punctate or strip-shaped erosion, or ulcers were observed. After administration of the positive drug, the number of strip-shaped hemorrhages on the gastric mucosa decreased, and no severe lesions were found.

Following the administration of *D. denneanum* crude polysaccharides, the number of punctate and strip-shaped hemorrhages on the gastric mucosa gradually decreased with increasing dosage. The gastric mucosal lesions in the high-dose *D. denneanum* crude polysaccharide groups (Groups G–I) were significantly improved compared to the low-dose groups (Groups D–F), with enhanced mucosal elasticity and increased folds.

As presented in Table 7, compared with the model group, the severity of ulcers in the positive drug group and all polysaccharide-administered groups was significantly reduced. The ulcer area in each high-dose polysaccharide group was significantly smaller than that in the corresponding low-dose group, and there was no significant difference between the high-dose polysaccharide groups and the positive drug group. Compared with the blank group, the ulcer area in the positive drug group and all polysaccharide-administered groups was significantly reduced, with the positive drug group showing the highest ulcer inhibition rate (93.16 ± 2.33). These results indicate that intragastric administration of *D. denneanum* crude polysaccharides (DDP) can significantly improve gastric mucosal ulcers in rats.

#### 3.1.7. Histopathological Analysis of Acute Gastric Ulcer Treated with *Dendrobium denneanum* Crude Polysaccharides

As depicted in Figure 5: Blank group (A): The gastric mucosa exhibited a complete tissue structure, with neatly arranged gastric mucosal epithelial cells showing normal morphology. No red blood cells or inflammatory cell infiltration were observed. Model group (B): The gastric mucosal tissue structure was severely damaged. Small-scale ulcers (black arrows) were visible in the gastric mucosal layer, with unclear structure in the ulcer area and slight epithelial cell detachment. Massive hemorrhage (green arrows) was observed, and a large number of lymphocytes and granulocytes infiltrated the deep mucosa (red arrows). The gastric glands and surrounding connective tissue were loosely arranged; mild dilation of gastric glands was occasionally observed (yellow arrows), with eosinophilic substances present in the glands. The connective tissue was loosely arranged, accompanied by the infiltration of a large number of lymphocytes, macrophages, and granulocytes (dark red arrows).Positive drug group: Small-scale ulcers (black arrows) were seen in the gastric mucosal layer, with unclear structure in the ulcer area, pyknosis, fragmentation, or lysis of cell nuclei, and slight hemorrhage (green arrows). A small number of lymphocytes and granulocytes infiltrated the deep mucosa (red arrows). The degree of gastric mucosal damage was significantly alleviated compared to the model group. Low-dose polysaccharide groups (D–F): Small-scale ulcers (black arrows) were visible in the gastric mucosal layer, with unclear ulcer structure, pyknosis, fragmentation, or lysis of cell nuclei, and moderate hemorrhage (green arrows). A small number of lymphocytes and granulocytes infiltrated the deep mucosa (red arrows); mild dilation of gastric glands was observed (yellow arrows), with eosinophilic substances in the glands and slight detachment of gastric gland epithelial cells (brown arrows). The submucosa showed extensive mild edema (light blue arrows), with loosely arranged connective tissue and infiltration of a small number of lymphocytes and granulocytes (dark red arrows). The overall severity of lesions was similar to that of the model group, but the ulcer area was significantly reduced. High-dose polysaccharide groups: Compared with the low-dose groups, the ulcer range was significantly reduced, and the integrity of the mucosal structure was partially restored. The infiltration of lymphocytes and granulocytes in the deep mucosa decreased (red arrows). Although red blood cells and inflammatory cells were still present in all high-dose groups, their quantity and severity were reduced (dark red arrows). High-dose administration significantly improved ethanol-induced diffuse hyperemia of the gastric mucosa and epithelial desquamation in rats with acute gastric ulcer. The mucosal repair effect showed a dose-dependent trend: the high-dose polysaccharide groups exhibited better effects than the low-dose groups, with significantly reduced epithelial cell detachment and decreased infiltration of lymphocytes and granulocytes.

Based on the results of gastric tissue morphological analysis in each group mentioned above, and evaluated in accordance with the criteria outlined in Table 2, the findings are presented in Table 8. Specifically, the Mucosal Edema score of the model group was 4, while that of all other groups was 0. For Hemorrhage, the model group achieved a score of 3, the blank group scored 0, and the groups treated with polysaccharides from different origins at various concentrations had a Hemorrhage score of 2, representing a decrease compared with the model group. Additionally, regarding Inflammatory-Cell Infiltration and Epithelial-Cell Loss, the scores of the high-dose polysaccharide groups (from the three origins) were lower than those of the low-dose polysaccharide groups (from the same three origins).

#### 3.1.8. Scanning Electron Microscopy (SEM) Analysis of Gastric Pits

As shown in Figure 6: Blank group (A): The gastric mucosal surface exhibited typical characteristics of a healthy structure. The gastric pits were intact, and the mucosal surface was composed of closely arranged columnar epithelial cells. Tight junctions were formed between adjacent epithelial cells, with clear cell boundaries and intact morphology. The mucus gel network had a dense, honeycomb-like porous structure, and no detachment or fragmentation was observed. Model group (B): The gastric pit structures were severely damaged, with extensive detachment of epithelial cells, forming ulcerative concave holes with a diameter of 20–30 μm and exposing the underlying connective tissue. The residual cells were swollen and deformed, with blurred boundaries and expanded intercellular spaces (2–3 μm). The function of mucus synthesis and secretion was severely impaired. Positive drug group (C): Mild ulcers were present, with ulcer diameters of 5–10 μm. The epithelial cells remained relatively intact, and partially damaged epithelial cells showed a significant recovery trend. The number of newly formed epithelial cells increased and were closely arranged, with a coverage rate > 90%.Low-dose polysaccharide groups (D–F): The gastric pit structures were severely damaged, with extensive epithelial cell detachment and the formation of ulcerative concave holes (10–20 μm in diameter); the underlying connective tissue was exposed in some areas. The residual cells were swollen and deformed, with blurred boundaries and expanded intercellular spaces (2–3 μm). The functions of mucus synthesis and secretion were impaired. The newly formed epithelial cells in the marginal area were loosely arranged, showing a reconstruction trend on the mucosal surface, but the intercellular spaces of epithelial cells remained large, indicating a severe degree of damage. High-dose polysaccharide groups (G–I): SEM results showed that the damage to gastric pits was significantly improved. The diameter of ulcerative concave holes in epithelial cells was reduced to less than 5 μm, and the epithelial cells showed a significant recovery trend with improved integrity. The number of newly formed epithelial cells increased. The SEM findings of the high-dose polysaccharide groups were similar to those of the positive drug group.

#### 3.1.9. Effects of *Dendrobium denneanum* Crude Polysaccharides on Oxidative Stress and Inflammation-Related Indices in Serum and Gastric Tissue

As presented in Table 9, compared with the blank group, the model group showed severe oxidative stress: the levels of glutathione peroxidase (GSH-Px), catalase (CAT), superoxide dismutase (SOD), and nitric oxide (NO) in serum and gastric tissue were significantly lower, while the level of malondialdehyde (MDA) was significantly higher. Compared with the blank group, the omeprazole (OME) group significantly alleviated oxidative damage in gastric tissue and serum, increased the levels of GSH-Px, CAT, SOD, and NO, and decreased the MDA content. The three low-dose polysaccharide groups increased the levels of GSH-Px, CAT, SOD, and NO in serum and gastric tissue, and decreased the MDA content to varying degrees; the high-dose polysaccharide groups showed significantly better damage-alleviating effects than the low-dose groups, presenting a dose-dependent trend.

As shown in Table 10, in the model group, the levels of inflammatory indices (tumor necrosis factor-α, TNF-α; interleukin-6, IL-6) in serum and gastric tissue were significantly higher than those in other groups, while the content of prostaglandin E2 (PGE2) in gastric tissue was significantly lower. Compared with the model group, the OME group showed significantly decreased TNF-α and IL-6 levels, but these levels were still lower than those in the blank group—indicating that the positive drug could only alleviate ethanol-induced damage rather than achieve a complete cure. Compared with the model group, the low-dose polysaccharide groups showed significantly decreased TNF-α and IL-6 levels and increased PGE2 content, but their TNF-α and IL-6 levels were still significantly higher than those in the PRE group. The high-dose polysaccharide groups exhibited significantly better damage-alleviating effects than the low-dose groups in a dose-dependent manner, and some indices were comparable to those in the PRE group. These results suggest that *Dendrobium denneanum* crude polysaccharides exert a favorable therapeutic effect on alcoholic acute gastric ulcer.

## 4. Discussion

This study systematically analyzed the content, structural characteristics, in vitro antioxidant activity, and gastric mucosal protective effect of *Dendrobium denneanum* polysaccharides (DDP) from different habitats. The core finding is that there are significant differences in total sugar content, molecular weight distribution, and monosaccharide composition of DDP from different habitats, and these structural characteristics are directly associated with their experimental activities. This provides a scientific basis for the precise development and utilization of DDP.

This study found that the total sugar content (20–51.49%) and yield (0.29–1.76%) of DDP from 22 habitats varied significantly, which may be closely related to local cultivation methods and environmental conditions [18]. In the field of Chinese medicinal materials, the regulatory effect of environmental factors on the accumulation of bioactive substances has been widely confirmed: for example, The results showed that the optimal conditions for the growth of *P. purpureum* were salinity 34 ppt, N:P ratio 169:1, and pH 8; for polysaccharide acquisition, they were salinity 17 ppt, N:P ratio 14:1, and pH 8; for phycoerythrin acquisition, salinity 17 ppt, N:P ratio 68:1, and pH 8; and for lipid acquisition, salinity 34 ppt, N:P ratio 1:1, and pH 8 [19]; Ecological factors such as annual mean precipitation and average temperature significantly affect the content of bioactive components in *Sinopodophyllum hexandrum* (Royle) T.S. Ying, with annual mean precipitation being the key determinant and showing a significant negative correlation with bioactive components. Notably, climatic factors contribute more to the variation than edaphic factors, and habitats like Jingyuan (Ningxia) are favorable for the accumulation of specific bioactive components [20]. Light intensity regulates the biosynthesis of terpenoid bioactive components in *Schizonepeta tenuifolia* Briq. via m^6^A RNA methylation: dark conditions increase menthone content by approximately 40-fold, while high light intensity promotes limonene accumulation [21]. In contrast, The microbial effects on monoterpene accumulation in citrus from the core region were further verified via synthetic community (SynCom) experiments. Rhizosphere microorganisms activated terpene synthesis and promoted monoterpene accumulation by interacting with the host immune system, while soil-derived endophytes with terpene synthesis potential might enhance citrus monoterpene accumulation by providing monoterpene precursors [22]. It is speculated that factors such as light intensity, precipitation distribution, soil fertility, and biotic interference in different habitats may regulate the activity of polysaccharide synthesis-related enzymes (e.g., UDP-glucosyltransferase) and metabolic pathways, ultimately leading to differences in DDP content. However, this study has not identified specific key environmental factors, which means the conclusions cannot yet directly guide the directional cultivation of DDP raw materials. In the future, it is necessary to combine detailed environmental data of each habitat (e.g., annual sunshine hours, soil pH) and use correlation analysis and controlled experiments (e.g., artificial regulation of light gradients) to further identify the core environmental factors affecting DDP accumulation, thereby achieving precise “environment-content” matching.

The biological activity of polysaccharides is determined by their structural characteristics. In this study, the structural data obtained via gel permeation chromatography (GPC), high-performance liquid chromatography (HPLC), and Fourier transform infrared spectroscopy (FT-IR) directly correspond to the in vitro antioxidant activity and gastric mucosal protective activity, clarifying the core mechanism of the “structure-function” relationship of DDP.

GPC results showed that the weight-average molecular weight (Mw) of DDP from 3 representative habitats was all less than 20 kDa. Among them, DDP5 (10.326 kDa) and DDP9 (10.991 kDa) were concentrated in the range of 10–11 kDa, while DDP14 (5.05 kDa) was a low-molecular-weight component. This is consistent with the research conclusion that low-molecular-weight polysaccharides have activity advantages: for instance, a comparison between two *Tremella fuciformis* acidic polysaccharides with different molecular weights, namely TFP (2238 kDa) and TFLP (molecular weight data to be supplemented), revealed that TFLP exhibited superior immunostimulatory and antioxidant activities compared to TFP [23]. Small-molecular-weight polysaccharide (DANP-II) was extracted from *Taraxacum mongolicum* via ultrasound-assisted enzymatic extraction (UAEE). Compared with other high-molecular-weight polysaccharides from *Taraxacum mongolicum*, DANP-II exhibited a more potent ability to alleviate H_2_O_2_-induced cell damage, reduce the cell apoptosis rate, and demonstrated strong antioxidant activity [24]. From the perspective of structural basis, the broad peak at 3386.55 cm^−1^ in the FT-IR spectrum confirmed that DDP is rich in hydroxyl groups (O-H). Low-molecular-weight DDP has smaller steric hindrance, making it easier to expose active hydroxyl groups. Therefore, although the in vitro antioxidant activity of DDP14 (with a low-molecular-weight characteristic of 5.05 kDa) was significantly lower than that of DDP5 and DDP9, its ulcer inhibition rate at low doses was higher than that of DDP5 and DDP9. It is speculated that DDP14 may penetrate gastric epithelial cell membranes to directly scavenge intracellular reactive oxygen species (ROS), which also explains why low-molecular-weight DDP exhibits better in vivo protective effects.

HPLC analysis revealed that DDP from the 3 habitats was mainly composed of glucose (Glc, 49.67–84.73%), mannose (Man, 8.29–12.25%), and galactose (Gal, 0.96–16.41%), but differences in their molar ratios directly affected activity. DDP5 had the highest Gal content (16.41%), with a Man:Glc:Gal molar ratio of 10.39:49.67:16.41. The characteristic peak of β-glycosidic bonds at 890 cm^−1^ in the FT-IR spectrum suggested that DDP5 may form a highly branched “comb-like” structure. This topological structure can delay free radical chain reactions through steric hindrance, which directly corresponds to the strongest hydroxyl radical scavenging activity of this sample (IC_50_ = 0.89 mg/mL). This result is consistent with the high antioxidant activity of the galactose-rich exopolysaccharide produced by *Limosilactobacillus fermentum* YL-11 [25].

The Man content of DDP9 (18.29%) was significantly higher than that of other samples. It is speculated that DDP9 may bind to immunoglobulin-like receptors (e.g., CD206) via the C-2 hydroxyl groups of Man residues [26], thereby assisting in enhancing anti-inflammatory activity. In the experiment, the TNF-α level in the high-dose DDP9 group (23.5 ± 2.1 pg/mL) was significantly lower than that in the DDP14 group (28.7 ± 3.4 pg/mL), which confirms this structural speculation. In contrast, DDP14 had a Gal content of only 0.96% and was mainly composed of linear glucans. Its in vitro antioxidant activity and in vivo anti-inflammatory activity were weaker than those of DDP5 and DDP9, which is consistent with the research conclusion that linear highland barley β-glucans have lower activity than branched structures [27]. In addition, the weak absorption peak near 670 cm^−1^ in the FT-IR spectrum suggested that DDP5 and DDP9 may contain small amounts of fucose and rhamnose. These special monosaccharides may further optimize the spatial conformation of polysaccharides, but their specific contributions need to be verified by subsequent purification of single polysaccharide components.

FT-IR analysis showed that the strong absorption peak of DDP at 3420 cm^−1^ confirmed the high abundance of hydroxyl groups, which is the basis for its antioxidant activity; the C-H stretching vibration peak at 2936.93 cm^−1^ indicated a stable pyranose ring structure; the weak absorption peak at 1240 cm^−1^ implied the possible presence of sulfate groups (S=O stretching vibration), while the absence of an obvious peak at 1730 cm^−1^ indicated low acetylation of DDP. Sulfate groups may promote electron transfer by enhancing the negative charge of polysaccharides, thereby improving activity. For example, The polysaccharide from *Sargassum carpophyllum* with the highest sulfated polysaccharide content exhibited the most potent effects in vitro: it inhibited nitric oxide (NO) production, reduced the release of cellular NO and reactive oxygen species (ROS), and suppressed the secretion of proinflammatory mediators in lipopolysaccharide (LPS)-stimulated RAW 264.7 macrophages [28]. The weak sulfation characteristic of DDP in this study may be an important supplement to its activity, but the specific effect needs to be further verified through sulfation modification experiments.

Based on experimental data, the protective effect of DDP against ethanol-induced acute gastric ulcers can be summarized as a “triple synergistic mechanism,” which is directly associated with structural characteristics: Antioxidant damage: Low-molecular-weight DDP (e.g., DDP14) can penetrate cell membranes, significantly increasing the activities of SOD (2.3 ± 0.2 U/mg prot) and CAT (1.8 ± 0.1 U/mg prot) in gastric tissue, while decreasing MDA content (1.5 ± 0.1 nmol/mg prot). This is consistent with its structural characteristic of sufficient hydroxyl group exposure. Anti-inflammatory regulation: DDP9 with high Man content can more effectively reduce the levels of TNF-α (23.5 ± 2.1 pg/mL) and IL-6 (35.2 ± 2.8 pg/mL), while increasing PGE_2_ content (120.3 ± 5.2 pg/mL). It is speculated that DDP9 may inhibit inflammatory pathways by regulating immune receptor signals. Mucosal repair: Scanning electron microscopy showed that the diameter of ulcerative concave holes in gastric pits was reduced to less than 5 μm in the high-dose DDP group, and the tight junctions of epithelial cells were restored. Combined with the β-glycosidic bond characteristic indicated by FT-IR, it is speculated that DDP may enhance barrier function by upregulating the expression of tight junction proteins (occludin, ZO-1), but this speculation has not been verified by molecular detection.

In addition, this study found that the approximately 3% total phenols in DDP may have a synergistic effect with polysaccharides: the phenolic hydroxyl groups of phenolic substances can form hydrogen bonds with the hydroxyl groups of polysaccharides, further enhancing free radical scavenging ability [29]; at the same time, phenols can inhibit myeloperoxidase activity and assist in reducing inflammatory infiltration [30]. This “polysaccharide-phenol” synergistic effect may be an important reason why the activity of DDP is superior to that of a single component, but the specific synergistic ratio and mode of action still need to be verified after separation and purification.

This study speculates that signaling pathways such as Nrf2/ARE and NF-κB may be involved in the protective effect of DDP, but direct experimental evidence is currently lacking. This is only a reasonable hypothesis based on existing results, not a definitive conclusion. In the future, it will be necessary to detect the expression levels of Nrf2, HO-1 (Nrf2 downstream target protein), and p65 (NF-κB subunit) via Western blot to verify whether these pathways are involved.

Based on the above limitations, future research can be carried out in three directions: Directional screening of high-quality raw materials: Combine environmental data (e.g., light, soil nutrients) of different habitats with DDP activity, use multiple regression analysis to identify key environmental factors, and establish a “habitat-structure-activity” prediction model to guide the standardized cultivation of high-quality DDP raw materials. Elucidation of specific mechanisms of action: Use Western blot and RT-PCR to detect the expression of Nrf2/ARE pathway-related genes (e.g., SOD1, CAT) and tight junction proteins (occludin, ZO-1), so as to clarify the molecular targets of DDP in regulating the gastric mucosal barrier. Advancement of preclinical translational research: Conduct in vivo pharmacokinetic studies of DDP (e.g., determination of half-life and bioavailability), and evaluate its safety through long-term administration experiments (e.g., 3-month gavage), so as to provide complete data support for the development of functional foods or candidate drugs.

In conclusion, this study is the first to clarify the structure-activity relationship of DDP from different habitats, confirming that DDP protects gastric mucosa through a “antioxidant-anti-inflammatory-mucosal repair” synergistic effect. Moreover, DDP with low molecular weight and high Gal/Man ratio exhibits better activity, which provides a core basis for the precise development of DDP.

## 5. Conclusions

This study compared the active component contents and in vitro/in vivo antioxidant activities of crude polysaccharides from *Dendrobium denneanum* collected from 22 habitats. The results showed that there were differences in the contents of active components among different habitats, which were speculated to be related to the cultivation environment and planting methods. These content differences also led to variations in antioxidant activity; however, all samples exhibited a certain degree of antioxidant capacity and exerted a preventive effect on acute alcoholic gastric ulcer. Animal experiments further confirmed that *D. denneanum* crude polysaccharides significantly improved the symptoms of gastric ulcer, alleviated tissue damage, and ameliorated the oxidative stress state by increasing the activity of antioxidant enzymes in serum and gastric tissue, thereby enhancing the gastric mucosal barrier. This study clarifies the gastric mucosal protective effect of *D. denneanum* crude polysaccharides, laying a theoretical foundation for the development of gastric protectants based on antioxidant mechanisms. However, its specific mechanism of action still requires further investigation.

## Figures and Tables

**Figure 1 foods-14-03258-f001:**
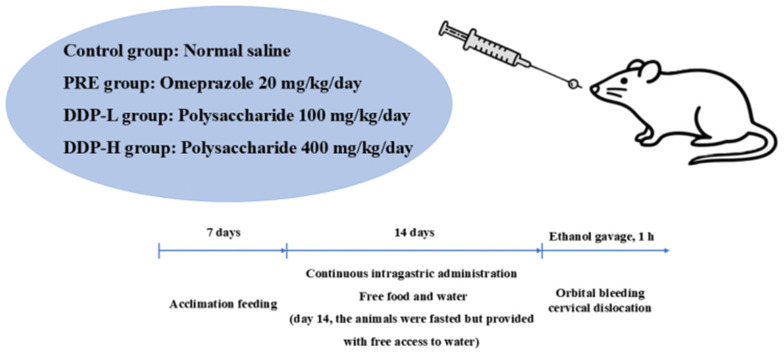
Flow Diagram for Acute Gastric Ulcer Rat Experiment.

**Figure 2 foods-14-03258-f002:**
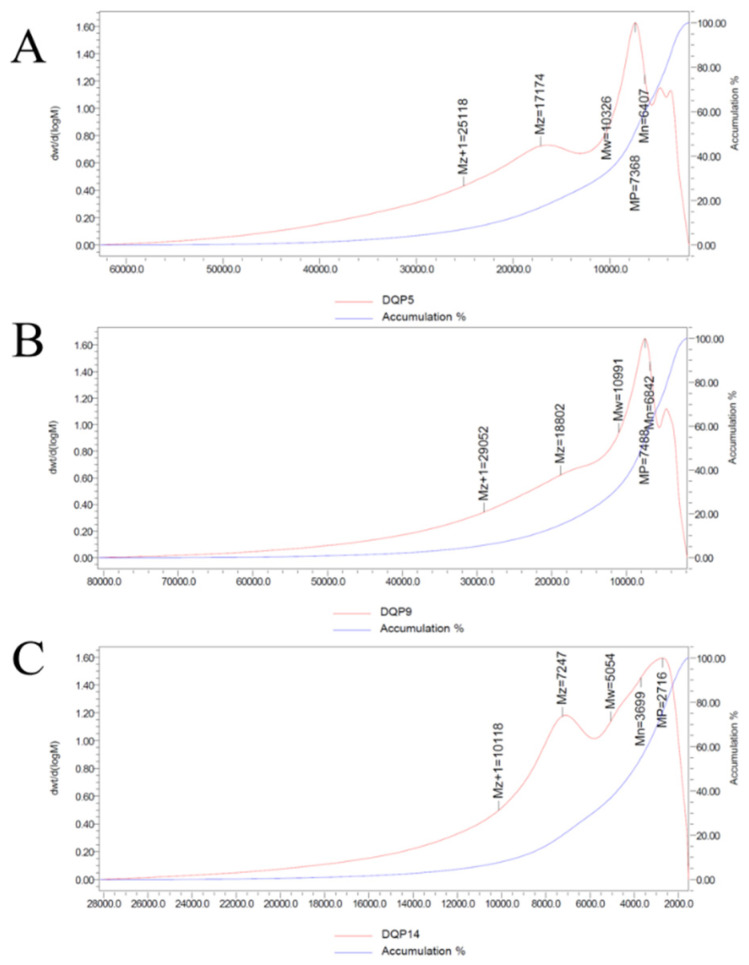
Molecular weights of DDP5 (**A**), DDP9 (**B**), and DDP14 (**C**).

**Figure 3 foods-14-03258-f003:**
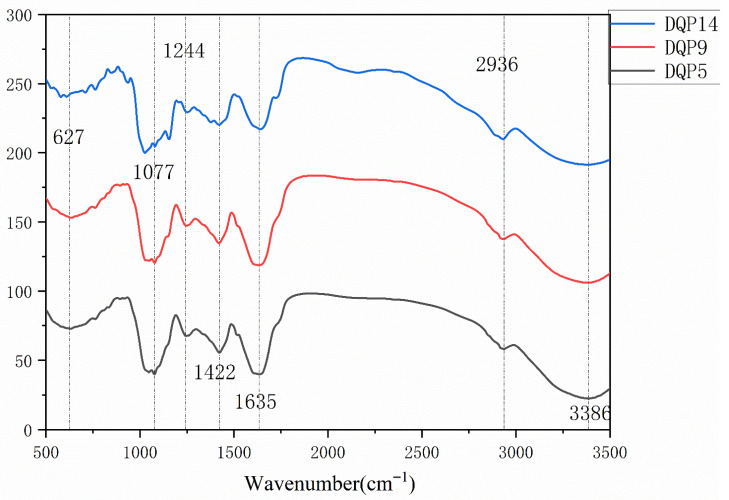
Fourier-transform infrared spectroscopic structure of crude polysaccharides from *Dendrobium denneanum* of different origins.

**Figure 4 foods-14-03258-f004:**
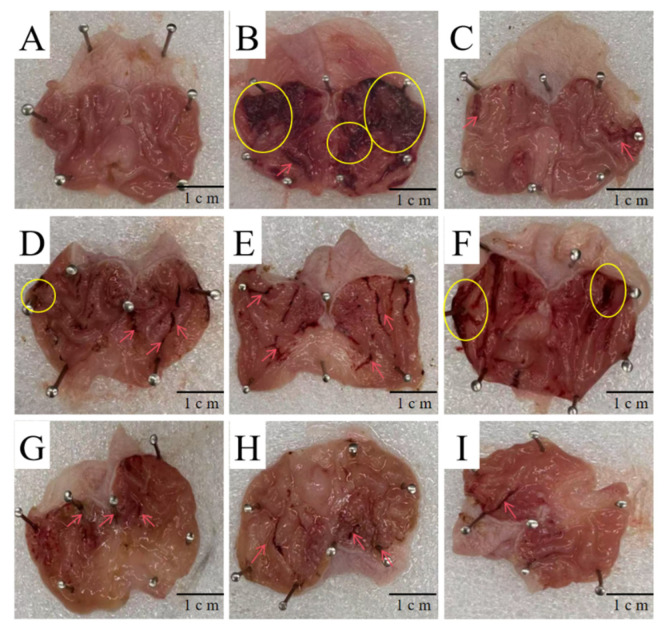
Macroscopic Observation of Gastric Mucosa((**A**): Blank group; (**B**): Model group; (**C**): PRE group; (**D**): DDP5L group; (**E**): DDP9L group; (**F**): DDP14L group; (**G**): DDP5H group; (**H**): DDP9H group; (**I**): DDP14H group. → indicates strip-shaped hemorrhage; ⚪ indicates severe lesion).

**Figure 5 foods-14-03258-f005:**
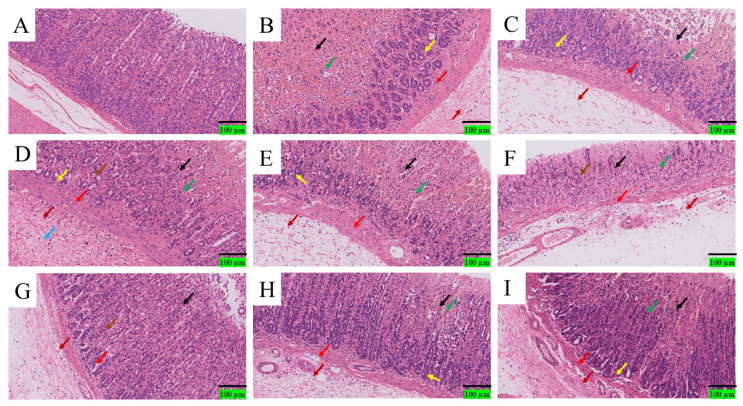
Histopathological Analysis (**A**): Blank group; (**B**): Model group; (**C**): PRE group; (**D**): DDP5L group; (**E**): DDP9L group; (**F**): DDP14L group; (**G**): DDP5H group; (**H**): DDP9H group; (**I**): DDP14H group. → (black arrow): Ulcer; → (green arrow): Hemorrhage; → (red arrow): Lymphocyte and granulocyte infiltration; → (brown arrow): Gastric gland epithelial cell detachment; → (dark red arrow): Loosely arranged connective tissue; → (light blue arrow): mild edema.

**Figure 6 foods-14-03258-f006:**
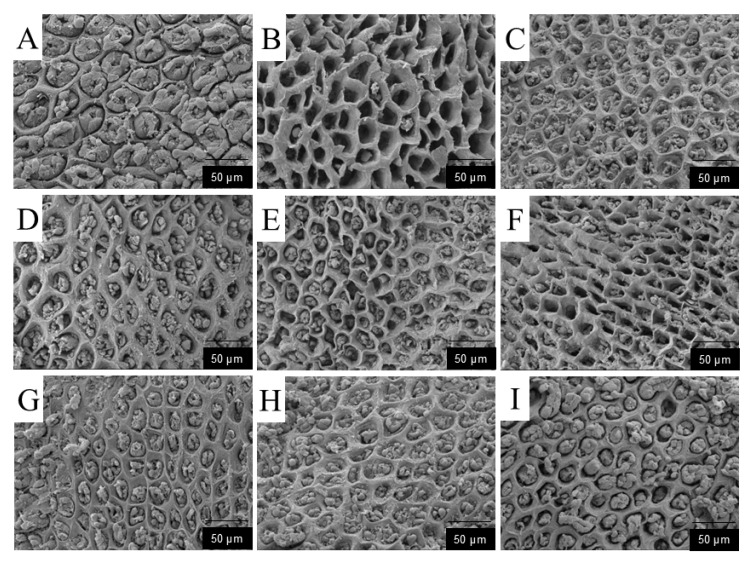
Scanning Electron Microscopic Analysis of Gastric Pits ((**A**): Blank group; (**B**): Model group; (**C**): PRE group; (**D**): DDP5L group; (**E**): DDP9L group; (**F**): DDP14L group; (**G**): DDP5H group; (**H**): DDP9H group; (**I**): DDP14H group).

**Table 1 foods-14-03258-t001:** Sample origin.

No.	Longitude (°E)	Latitude (°N)	Elevation (m)	Origin
1	98.324	27.221	1331.7	Mengzhong Village, Xieluo Township, Shimian County, Ya’an City, Sichuan, China
2	103.336	29.667	1300.5	Xinlin Group 4, Qiliping Town, Taoyuan County, Meishan City, Sichuan, China
3	103.765	29.552	721.1	Chuanxi Community, Huatou Town, Jiajiang County, Leshan City, Sichuan, China
4	102.468	29.242	1200.3	Xinle Village, Xieluo Township, Shimian County, Ya’an City, Sichuan, China
5	114.654	29.242	1300.9	Hualin Group 4, Qiliping Town, Taoyuan County, Meishan City, Sichuan, China
6	103.355	29.242	621.2	Liuxi Village, Maliu Township, Jiajiang County, Leshan City, Sichuan, China
7	103.574	29.242	548.4	Zhengjie Village, Huatou Town, Jiajiang County, Leshan City, Sichuan, China
8	102.476	29.242	1311.7	Dalin Village, Yonghe Township, Shimian County, Ya’an City, Sichuan, China
9	103.281	29.242	810.6	Tongsheng Community, Zhongbao Town, Hongya County, Meishan City, Sichuan, China
11	102.283	29.242	900.2	Lashu Village, Fengle Township, Shimian County, Ya’an City, Sichuan, China
12	103.574	29.242	632.8	Qianfeng Village, Maliu Township, Jiajiang County, Leshan City, Sichuan, China
13	103.285	29.242	800.8	Qiaokou Village, Dongyue Town, Hongya County, Meishan City, Sichuan, China
14	103.571	29.242	700.3	Lianhe Xincun Group 4, Xiema Township, Jiajiang County, Leshan City, Sichuan, China
15	102.22	29.242	920.1	Jiefang Village, Anshun Town, Shimian County, Ya’an City, Sichuan, China
16	100.322	29.242	1320	Dawan Village Group 3, Xieluo Township, Shimian County, Ya’an City, Sichuan, China
17	103.571	29.242	621.4	Fuxing Village, Xiema Township, Jiajiang County, Leshan City, Sichuan, China
18	102.468	29.242	1353.2	Gaoqiao Village, Fengle Township, Shimian County, Ya’an City, Sichuan, China
19	106.053	29.242	645.6	Yuanmen Village, Maliu Township, Jiajiang County, Leshan City, Sichuan, China
20	106.231	29.242	722.9	Yongxin Group 7, Huatou Town, Jiajiang County, Leshan City, Sichuan, China
21	102.345	29.242	1010.2	Jinping Village, Anshun Town, Shimian County, Ya’an City, Sichuan, China
22	103.21	29.242	800.6	Zhonghe Village, Gaomiao Town, Hongya County, Meishan City, Sichuan, China

**Table 2 foods-14-03258-t002:** Scoring criteria for gastric histological injury.

Score	Mucosal Edema	Hemorrhage	Inflammatory-Cell Infiltration	Epithelial-Cell Loss
0	No lesion	No lesion	No infiltration	No ulcer
1	Mild edema	Congestion	Superficial mucosal infiltration	A few small ulcers
2	Mucosal edema	Petechial hemorrhage	Marked mucosal infiltration	Large ulcers
3	Mucosal and submucosal micro-edema	Micro-hemorrhagic foci	Mucosal and submucosal micro-infiltration	Ulcers throughout the stomach
4	Severe mucosal and submucosal edema	Extensive hemorrhage	Severe mucosal and submucosal infiltration	Perforation of the stomach

**Table 3 foods-14-03258-t003:** Determination of Crude Polysaccharide Content in *Dendrobium denneanum* from Different Producing Areas.

No.	Total Sugar (%)	Total Phenolics (%)	Crude Polysaccharide Yield (%)
1	27.38 ± 2.73 c	3.11 ± 0.24 ab	0.58 ± 0.04 c
2	22.12 ± 0.75 c	2.64 ± 0.09 d	0.69 ± 0.07 bc
3	27.63 ± 2.42 c	2.89 ± 0.07 c	0.39 ± 0.05 e
4	27.97 ± 2.08 c	2.86 ± 0.11 c	0.43 ± 0.02 d
5	26.59 ± 0.66 c	3.58 ± 0.25 a	0.31 ± 0.04 e
6	29.26 ± 2.67 bc	3.04 ± 0.10 b	0.70 ± 0.05 b
7	30.16 ± 1.90 bc	3.21 ± 0.08 ab	1.76 ± 0.08 a
8	29.01 ± 3.49 c	3.23 ± 0.02 a	0.47 ± 0.10 d
9	20.23 ± 0.09 d	2.68 ± 0.07 d	0.37 ± 0.08 e
10	51.49 ± 2.94 a	2.74 ± 0.05 c	0.93 ± 0.03 b
11	25.50 ± 0.59 cd	3.05 ± 0.10 b	0.49 ± 0.09 d
12	38.66 ± 2.67 b	3.12 ± 0.06 ab	0.35 ± 0.09 e
13	44.25 ± 1.80 a	3.49 ± 0.30 a	0.45 ± 0.03 d
14	29.62 ± 2.84 bc	3.06 ± 0.09 b	0.41 ± 0.04 d
15	27.53 ± 3.56 cd	3.01 ± 0.20 bc	0.29 ± 0.07 e
16	33.18 ± 2.15 bc	2.87 ± 0.03 c	1.04 ± 0.07 b
17	33.73 ± 1.74 bc	3.03 ± 0.03 b	0.42 ± 0.03 d
18	23.91 ± 1.00 cd	3.22 ± 0.12 ab	0.84 ± 0.03 b
19	31.54 ± 2.86 bc	3.01 ± 0.17 bc	0.37 ± 0.02 e
20	24.79 ± 2.84 cd	3.10 ± 0.15 b	0.33 ± 0.01 e
21	23.97 ± 0.80 cd	2.73 ± 0.09 d	0.54 ± 0.06 c
22	26.33 ± 1.19 cd	3.37 ± 0.36 a	0.66 ± 0.02 bc

Different letters in each column indicate significant differences at *p* < 0.05, *n* = 3.

**Table 4 foods-14-03258-t004:** Determination of in vitro antioxidant capacity of crude polysaccharides from *Dendrobium denneanum* of different origins.

No.	ABTS Radical Scavenging IC_50_ Value (mg/mL)	DPPH Radical Scavenging IC_50_ Value (mg/mL)	Hydroxyl Radical Scavenging IC_50_ Value (mg/mL)
1	1.18 ± 0.17 c	1.37 ± 0.09 bc	1.44 ± 0.13 b
2	1.38 ± 0.13 b	1.67 ± 0.08 b	1.69 ± 0.16 b
3	1.09 ± 0.11 c	1.23 ± 0.08 c	1.39 ± 0.08 b
4	1.62 ± 0.16 a	2.13 ± 0.19 a	2.20 ± 0.07 a
5	0.61 ± 0.18 d	0.99 ± 0.04 d	1.02 ± 0.18 c
6	1.01 ± 0.07 c	1.21 ± 0.08 c	1.35 ± 0.13 b
7	0.62 ± 0.11 d	1.01 ± 0.06 d	1.04 ± 0.08 c
8	1.22 ± 0.13 bc	1.52 ± 0.13 b	1.48 ± 0.09 b
9	1.52 ± 0.12 a	2.11 ± 0.16 a	2.18 ± 0.09 a
10	1.41 ± 0.13 b	1.75 ± 0.17 b	1.71 ± 0.12 b
11	0.99 ± 0.02 c	1.21 ± 0.03 c	1.34 ± 0.09 b
12	0.97 ± 0.05 c	1.08 ± 0.12 d	1.09 ± 0.08 c
13	1.18 ± 0.08 c	1.27 ± 0.09 c	1.43 ± 0.08 b
14	1.56 ± 0.15 a	2.08 ± 0.11 a	2.04 ± 0.13 a
15	0.96 ± 0.08 c	1.03 ± 0.05 d	1.09 ± 0.07 c
16	1.42 ± 0.15 b	2.08 ± 0.11 a	2.08 ± 0.08 a
17	1.36 ± 0.12 b	1.62 ± 0.13 b	1.68 ± 0.19 b
18	1.35 ± 0.07 b	1.59 ± 0.08 b	1.51 ± 0.16 b
19	0.98 ± 0.14 c	1.11 ± 0.09 d	1.22 ± 0.09 bc
20	0.93 ± 0.09 c	1.02 ± 0.12 d	1.04 ± 0.12 c
21	1.33 ± 0.07 b	1.56 ± 0.08 b	1.51 ± 0.08 b
22	1.02 ± 0.05 c	1.22 ± 0.08 c	1.35 ± 0.18 b

Different letters in each column indicate significant differences at *p* < 0.05, *n* = 3.

**Table 5 foods-14-03258-t005:** Molecular weight of crude polysaccharides from *Dendrobium denneanum* of different origins.

Sample	MP	Mw	Mn	Polydispersity
DDP1	7368	10,326	6407	1.611733
DDP12	7488	10,991	6842	1.606471
DDP22	2716	5054	3699	1.366575

**Table 6 foods-14-03258-t006:** Monosaccharide composition of crude polysaccharides from *Dendrobium denneanum* of different origins.

Monosaccharide	DDP5		DDP9		DDP14	
	mol %	RT (min)	mol %	RT (min)	mol %	RT (min)
Man	10.39	12.63	18.29	12.55	12.25	12.58
Rib	2.1	16.42	1.88	16.31	0.13	16.37
Rha	3.9	17.16	3.09	17.05	0.32	17.11
GlcA	3	20.58	1.15	20.4	0.15	20.44
GalA	1.1	21.9	0.58	21.86	0.44	21.5
Glc	49.67	26.93	53.62	26.72	84.73	26.65
Gal	16.41	30.85	11.26	30.62	0.96	30.76
Xyl	3.22	32.57	2.72	32.34	0.42	32.46
Ara	9.1	33.76	6.54	33.51	0.62	33.67
Fuc	1	38.86	0.89	38.56	0	0

**Table 7 foods-14-03258-t007:** Ulcer index analysis.

Group	Ulcer Area Percentage (%)	Ulcer Area Inhibition Rate (%)
Control	0 ± 0 d	100 ± 0 a
Model	19.89 ± 1.32 a	0 ± 0 c
PRE	2.31 ± 0.18 c	93.16 ± 2.33 a
DDP5L	14.87 ± 1.47 b	48.47 ± 1.48 c
DDP5H	6.12 ± 1.02 c	81.62 ± 1.56 a
DDP9L	13.92 ± 2.27 b	46.91 ± 1.28 c
DDP9H	4.86 ± 0.89 c	82.47 ± 2.44 a
DDP14L	12.15 ± 1.77 b	48.72 ± 1.72 c
DDP14H	2.12 ± 0.15 c	69.26 ± 1.09 b

Different letters in each column indicate significant differences at *p* < 0.05, *n* = 5.

**Table 8 foods-14-03258-t008:** Scoring criteria for gastric histological injury.

Group	Mucosal Edema	Hemorrhage	Inflammatory-Cell Infiltration	Epithelial-Cell Loss
Control	0	0	0	0
Model	4	3	3	2
OME	0	2	3	1
DDP5L	0	2	3	1
DDP5H	0	2	2	0
DDP9L	0	2	3	1
DDP9H	0	2	2	0
DDP14L	0	2	3	1
DDP14H	0	2	2	0

**Table 9 foods-14-03258-t009:** Effects of crude polysaccharides from *Dendrobium denneanum* on oxidative stress in serum and gastric tissue.

Group	Stomach					Serum				
	CAT (U/mL)	SOD (U/mL)	MDA (nmol/mL)	GSH-PX (U/mL)	NO (μmol/g prot)	CAT (U/mL)	SOD (U/mL)	MDA (nmol/mL)	GSH-PX (U/mL)	NO (μmol/g prot)
Control	36.82 ± 7.14 b	343.41 ± 27.51 a	0.69 ± 0.09 c	106.48 ± 10.00 a	0.075 ± 0.005 b	443.63 ± 6.82 a	14.50 ± 1.90 b	1.34 ± 0.49 d	28.72 ± 0.50 a	0.352 ± 0.050 b
Model	15.28 ± 1.59 d	126.74 ± 26.53 d	1.63 ± 0.21 a	49.02 ± 14.65 c	0.039 ± 0.009 c	105.89 ± 16.19 e	11.14 ± 1.70 c	11.07 ± 1.09 a	5.70 ± 1.51 d	0.207 ± 0.099 d
PRE	44.46 ± 2.88 a	328.92 ± 32.07 a	0.79 ± 0.16 b	116.75 ± 8.83 a	0.082 ± 0.009 a	357.31 ± 36.06 b	12.13 ± 1.56 c	1.76 ± 0.44 d	20.72 ± 0.50 b	0.363 ± 0.089 b
DDP5L	34.50 ± 3.54 bc	278.56 ± 31.18 b	1.12 ± 0.04 b	84.73 ± 5.11 b	0.042 ± 0.013 c	189.33 ± 52.16 d	13.64 ± 1.01 b	1.90 ± 0.49 d	7.22 ± 1.00 c	0.252 ± 0.038 c
DDP5H	42.59 ± 2.35 a	370.48 ± 6.97 a	0.83 ± 0.06 b	118.11 ± 8.15 a	0.112 ± 0.016 a	266.78 ± 22.82 c	15.80 ± 1.75 a	1.18 ± 0.51 d	8.39 ± 0.81 c	0.336 ± 0.064 b
DDP9L	33.04 ± 1.56 c	248.14 ± 13.86 c	1.07 ± 0.16 b	76.86 ± 4.57 b	0.045 ± 0.005 c	233.41 ± 8.81 c	14.60 ± 1.19 b	5.07 ± 0.79 b	10.22 ± 1.00 c	0.327 ± 0.056 b
DDP9H	41.99 ± 2.85 a	321.96 ± 17.50 a	0.79 ± 0.12 b	116.53 ± 9.02 a	0.098 ± 0.014 a	265.68 ± 12.81 c	15.67 ± 1.36 a	2.57 ± 1.25 c	16.22 ± 1.00 b	0.451 ± 0.044 a
DDP14L	29.08 ± 6.02 c	272.18 ± 28.05 b	1.51 ± 0.27 a	88.45 ± 4.78 b	0.048 ± 0.009 c	102.95 ± 6.79 e	14.41 ± 1.77 b	3.95 ± 1.37 b	8.32 ± 0.47 c	0.179 ± 0.041 d
DDP14H	36.53 ± 5.35 b	307.51 ± 23.32 b	1.23 ± 0.30 ab	105.17 ± 10.89 a	0.058 ± 0.007 c	296.79 ± 50.98 bc	15.83 ± 0.76 a	1.43 ± 0.25 d	18.72 ± 0.50 b	0.311 ± 0.094 b

Different letters in each column indicate significant differences at *p* < 0.05, *n* = 5.

**Table 10 foods-14-03258-t010:** Effects of crude polysaccharides from Dendrobium denneanum on inflammation-related indicators in serum and gastric tissue.

Group	Stomach			Serum	
	IL-6 (pg/mL)	TNF-α (pg/mL)	PGE_2_ (pg/mL)	IL-6 (pg/mL)	TNF-α (pg/mL)
Control	83.63 ± 3.03 c	22.95 ± 0.52 d	75.90 ± 8.80 b	42.95 ± 0.52 c	16.29 ± 5.31 d
Model	142.85 ± 8.56 a	114.41 ± 13.16 a	38.06 ± 2.37 e	71.08 ± 2.15 a	57.74 ± 10.02 a
OME	88.43 ± 3.44 c	51.06 ± 1.33 bc	85.09 ± 4.02 a	47.72 ± 4.39 b	24.39 ± 2.62 c
DQP5L	113.76 ± 9.91 b	76.37 ± 10.11 b	73.15 ± 3.10 b	53.04 ± 2.38 b	33.04 ± 2.38 b
DQP5H	86.20 ± 6.69 c	34.92 ± 0.64 c	36.12 ± 2.29 e	44.92 ± 0.25 c	24.92 ± 0.25 c
DQP9L	101.96 ± 6.89 b	80.12 ± 1.66 b	44.21 ± 5.76 d	53.45 ± 4.27 b	33.45 ± 2.27 b
DQP9H	85.26 ± 10.61 d	43.51 ± 2.89 c	41.70 ± 10.39 d	43.51 ± 3.20 c	23.51 ± 3.20 c
DQP14L	104.10 ± 8.01 b	79.32 ± 9.54 b	73.63 ± 6.63 b	59.32 ± 1.49 b	35.98 ± 2.66 b
DQP14H	84.64 ± 8.60 c	48.93 ± 2.66 c	40.32 ± 10.39 d	42.27 ± 2.90 c	22.27 ± 2.90 c

Different letters in each column indicate significant differences at *p* < 0.05, *n* = 5.

## Data Availability

The original contributions presented in the study are included in the article, further inquiries can be directed to the corresponding authors.

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
