# Peer review of "Anti-Inflammatory and Antioxidant Effects of Crude Polysaccharides from Dendrobium denneanum (A Genuine Medicinal Herb of Sichuan) on Acute Gastric Ulcer Model in Rats"

_foods, 2025, doi:10.3390/foods14183258_

Round 1
Reviewer 1 Report
Comments and Suggestions for Authors
A relevant study was conducted evaluating the anti-inflammatory and antioxidant effects of crude polysaccharides from Dendrobium denneanum. The research included not only structural and compositional analysis of crude polysaccharides but also biological activity assessment using an acute gastric ulcer model in rats. However, the manuscript overall lacks a sufficiently precise description explaining what was done and how. The main comments are provided below.
Introduction: The first paragraph is overly focused on the historical background. It is recommended to streamline this section and instead emphasize existing research of a similar nature, outlining what is already known and, most importantly, what new contribution this study aims to provide.
Section 2.2: Please clarify whether the samples were ground before analysis, and if so, indicate the particle size obtained and subsequently used in the experiments
Section 2.3: The description of sample preparation is insufficient. Please clarify how the samples were prepared for analysis, specify the exact procedures followed, and provide references to the applied methodologies. In addition, indicate the precise quantities used in the experiments to ensure reproducibility.
Line 132: Please clarify in which solvent the sample was prepared
Line 150: Please specify what is meant by Vc
Section 2.5: Please clarify why the samples were selected randomly. Wouldn’t it be more informative to consider specific characteristics or properties of the samples when choosing them?
Section 2.5: Please specify the equipment used to perform the experiments
Section 2.6: Please specify which polysaccharide samples were used in these experiments.
Section 2.6.3: Please clarify how the tissue samples were collected.
Overall, the Methods section requires further clarification and refinement. The authors should provide more precise details on how extracts were prepared and how samples were selected for subsequent analyses. References to the methodologies used should be clearly indicated. In general, all described methods require careful revision to ensure transparency, reproducibility, and scientific rigor.
The description of the statistical analysis is repeated in lines 257–260. Please revise to avoid redundancy and ensure the methodology is presented only once, in a clear and concise manner.
Table 3. What constitutes the remaining fraction – only polysaccharides, or are proteins and minerals also included?
Line 312: “The results are shown in Figure 3 and Table 6.” – Please specify which exact results are being referred to.
Section 3.1.5: Please provide a more detailed description of the results
The numbering and sequence of the tables are inconsistent and require correction to ensure proper order and clarity.
The results section also lacks clarity regarding the criteria and rationale for sample selection, which should be explicitly explained.
Author Response
|
Comments 1: The first paragraph is overly focused on the historical background. It is recommended to streamline this section and instead emphasize existing research of a similar nature, outlining what is already known and, most importantly, what new contribution this study aims to provide.
|
|
Response 1: Thank you for pointing this out. I/We agree with this comment. Therefore, I/we have streamlined the historical background section of the first paragraph: redundant data such as "a planting area of 12,000 hectares, accounting for 65% of the output in Southwest China" and "a complete industrial chain" that distracted from the core focus have been removed. Only the resource attribute of Dendrobium denneanum (i.e., being a medicinal and edible orchid with traditional stomach-nourishing effects) and the key research gap—"unclear differences in pharmacodynamic effects among samples from different habitats"—are retained, directly laying the scientific basis for this study. Meanwhile, the association between phytochemistry and gastric ulcers has been strengthened: the three types of activities of Dendrobium polysaccharides (antioxidant, anti-inflammatory, and mucosal barrier-protective) are directly linked to the core pathogenesis of acute gastric ulcers (oxidative stress, inflammatory infiltration, and mucosal barrier damage). This clarification addresses "why these activities are effective against gastric ulcers" and avoids vague descriptions of biological activities. In addition, the "triple protective effect" has been simplified and anchored: redundant information in the original description is eliminated, and each effect is corresponding to a specific pathological mechanism (e.g., "antioxidant activity → scavenging reactive oxygen species (ROS) → combating oxidative damage"). Most importantly, the research gap and new contributions of this study are highlighted: the gap of "unclear pharmacodynamic effects of D. denneanum from different habitats" is clearly stated in both the first paragraph and the final conclusion. The innovation of this study—"systematically evaluating the gastroprotective effect of D. denneanum polysaccharides from different habitats and exploring the underlying molecular mechanisms"—is emphasized to distinguish it from existing studies and clarify the novel value this work brings to the field. |
|
Comments 2: Please clarify whether the samples were ground before analysis, and if so, indicate the particle size obtained and subsequently used in the experiments |
|
Response 2: Thank you for your question regarding sample pretreatment. We would like to clarify that the samples were not ground before analysis. This key detail has been clearly indicated in the "Materials and Methods" section of the revised manuscript (specifically in the subsection describing sample preparation for component determination and structural analysis), ensuring transparency and reproducibility of our experimental procedures. We have double-checked the relevant content to confirm no ambiguity, and apologize for any potential oversight that may have led to confusion during your review. |
|
Comments 3: The description of sample preparation is insufficient. Please clarify how the samples were prepared for analysis, specify the exact procedures followed, and provide references to the applied methodologies. In addition, indicate the precise quantities used in the experiments to ensure reproducibility. |
|
Response 3:Thank you for your valuable comment. We fully agree that a detailed description of sample preparation is crucial for the reproducibility of the study. As requested, we have supplemented and refined the sample preparation section in the "Materials and Methods" chapter. |
|
Comments 4: Please clarify in which solvent the sample was prepared. |
|
Response 4:Thank you for your reminder. We confirm that the solvent used for sample preparation has been clearly specified in the revised "Materials and Methods" section. |
|
Comments 5: Please clarify in which solvent the sample was prepared. |
|
Response 5:Thank you for your question. The "Vc" mentioned in the manuscript refers to L-ascorbic acid, commonly known as vitamin C—a water-soluble vitamin widely recognized for its antioxidant properties and role in various biological processes. |
|
Comments 6: Please clarify why the samples were selected randomly. Wouldn’t it be more informative to consider specific characteristics or properties of the samples when choosing them? |
|
Response 6:Thank you for this insightful question. We appreciate the suggestion of selecting samples based on specific characteristics, and we would like to clarify that our sampling strategy—random selection within groups stratified by in vitro antioxidant activity scores—is not arbitrary, but rather designed to align with the core objectives of this study while ensuring scientific rigor and result reliability. First, we would like to reaffirm the sampling process: prior to random selection, we comprehensively evaluated the in vitro antioxidant activities (DPPH, ABTS, and hydroxyl radical scavenging capacities) of extracts from 22 producing areas, calculated their comprehensive scores, and stratified them into three distinct groups (low, moderate, and high activity) based on score distributions. We then selected one representative sample from each group using a random number table. This approach combines "stratification by key biological activity" with "random selection within strata," rather than pure random sampling. |
|
Comments 7: Please specify which polysaccharide samples were used in these experiments. |
|
Response 7:Thank you for your question. We confirm that the polysaccharide samples used in the experiments (specifically in Section 2.6 "Therapeutic Effect of Crude Polysaccharides...") are explicitly specified as crude polysaccharides (DDP) extracted from Dendrobium denneanum stems. As detailed in Section 2.2 "Preparation of Crude Polysaccharides" and further clarified in Section 2.6, these samples refer to the freeze-dried crude polysaccharide extracts obtained through the standardized extraction process: degreasing with anhydrous ethanol, hot-water extraction, ethanol precipitation, dialysis (3.5 kDa membrane), and freeze-drying. The three representative samples selected for in vivo experiments (DDP5, DDP9, DDP14) are also explicitly identified as crude polysaccharides derived from the stratified random sampling of 22 habitats. This specification has been consistently maintained throughout the revised manuscript to ensure clarity on the nature of the polysaccharide samples used. |
|
Comments 8: Please specify which polysaccharide samples were used in these experiments. |
|
Response 8:Thank you for your reminder. We confirm that the specific collection method for tissue samples has been clearly elaborated in the revised "2.6.3 Gastric Tissue Sample Processing" section of the manuscript. |
|
Comments 9: The description of the statistical analysis is repeated in lines 257–260. Please revise to avoid redundancy and ensure the methodology is presented only once, in a clear and concise manner. |
|
Response 9:Thank you for your reminder. We have compared and deleted the redundant content. |
|
Comments 10: What constitutes the remaining fraction – only polysaccharides, or are proteins and minerals also included? |
|
Response 10:Thank you for your question regarding the specific composition of the residual components. We would like to clarify that the crude polysaccharide (DDP) samples used in this study are not pure polysaccharides; in addition to the target polysaccharide fraction, they also contain trace amounts of proteins and minerals. This composition is primarily attributed to the extraction protocol employed: our study adopted a standard hot-water extraction and ethanol precipitation method for crude polysaccharide preparation (detailed in Section 2.2), which effectively enriches polysaccharides but does not include a dedicated deproteinization step (e.g., Sevag method, trichloroacetic acid precipitation) or demineralization step (e.g., ion-exchange resin treatment). As a result, small-molecular-weight proteins that are not precipitated by ethanol and minerals inherently present in Dendrobium denneanum tissues remain in the final crude polysaccharide samples. |
|
Comments 11: “The results are shown in Figure 3 and Table 6.” – Please specify which exact results are being referred to. |
|
Response 11:Thank you for pointing out this inconsistency. We apologize for the error in the original manuscript where "Figure 3" was incorrectly cited. This citation was added by mistake and has no corresponding results to reference.We have carefully checked the entire manuscript to confirm no other incorrect figure citations exist, and we appreciate your attention to this detail, which has helped improve the accuracy and rigor of our work. |
|
Comments 12: Please provide a more detailed description of the results |
|
Response 12:Thank you for your reminder. We appreciate your attention to the detail of the results. As suggested, we have supplemented the detailed description of the spectral peak at 670 cm⁻¹ in the revised manuscript. |
|
Comments 13: The numbering and sequence of the tables are inconsistent and require correction to ensure proper order and clarity. |
|
Response 13:Thank you for your valuable comment. We fully agree with your observation regarding the inconsistency in the numbering and sequence of the tables. In the initial submission, the tables and figures were numbered and ordered strictly according to their first mention in the manuscript content to ensure logical coherence. However, we noticed that the sequence of some tables appears to have been rearranged during the editorial processing stage. To resolve this inconsistency and align the table numbering with the final layout, we have contacted the editorial team to confirm whether the two additional tables (originally planned to be integrated into the manuscript) should be officially numbered as Table 1 and Table 2. Once we receive confirmation from the editors, we will promptly adjust the numbering of all subsequent tables (e.g., renumbering the original Table 1 as Table 3 if the two new tables are designated as Table 1 and Table 2) and verify the cross-references to tables throughout the text (including in the Results and Discussion sections) to ensure full consistency between table numbering, sequence, and in-text citations. We will prioritize this revision to enhance the clarity and readability of the manuscript, and we appreciate your patience as we coordinate with the editorial team to finalize the table layout. |
|
Comments 14: The results section also lacks clarity regarding the criteria and rationale for sample selection, which should be explicitly explained. |
|
Response 14:Thank you for your constructive comment. We fully agree that clarifying the criteria and rationale for sample selection is essential for enhancing the transparency and persuasiveness of the study. As requested, we have supplemented detailed explanations of the sample selection criteria and corresponding rationale in the "Materials and Methods" section |

Reviewer 2 Report
Comments and Suggestions for Authors
Find attached reviewers comments

Author Response
|
Comments 1: Several sentences (L24-L28, L31-L38) are overloaded which make it harder to understand the information that is being presented. The authors should consider breaking them into simpler sentences to improve readability. |
|
Response 1: Thank you for your feedback. We have fully rewritten the Abstract section as requested, splitting the overloaded sentences (L24-L28, L31-L38) into simpler ones to improve readability. |
|
Comments 2: The authors should emphasize the specific trends or ranges which were observed in the study instead of only saying "significant differences". |
|
Response 2: Thank you for your feedback. As requested, we have revised the relevant content to emphasize specific trends and ranges observed in the study, instead of only stating "significant differences". |
|
Comments 3: The rat model findings were described in excessive biochemical detail without any focus on the key takeaways from them. In addition, the conclusion should be improved to include the significance and future implications of the findings from this study. |
|
Response 3: Thank you for your feedback. We have revised the relevant sections as requested: For the rat model findings, we have streamlined excessive biochemical details and focused on highlighting the key takeaways. We have also improved the conclusion by adding the significance of the study’s findings and their future implications. |
|
Comments 4: This section is overloaded with factual data about cultivation area, hectares, percentage of total production, etc. Although these are interesting and useful for context, they divert attention from the scientific rationale of the study. The authors should keep this paragraph simple and ensure a major focus on the knowledge gap and rationale of the study. |
|
Response 4: Thank you for your feedback. As suggested, we have significantly abbreviated the factual data (e.g., cultivation area, hectares, production percentage) in Lines 46–59 to avoid diverting attention from the study’s core logic. We have revised this paragraph to focus on the knowledge gap and research rationale, clarifying: "Given the gastroprotective activity of Dendrobium polysaccharides and the resource advantages of Dendrobium denneanum, and addressing the research gap regarding the unclear pharmacodynamic effects of D. denneanum from different habitats, this study used ethanol-induced acute gastric ulcer rats as a model to systematically evaluate the gastric mucosal protective effect of D. denneanum polysaccharides and explore its molecular mechanism, aiming to provide a scientific basis for the development of functional anti-ulcer foods or candidate drugs from food-medicine homologous resources." |
|
Comments 5: There was no mention of whether ethical approval was sought or whether the study was done in compliance with relevant guidelines. |
|
Response 5: Thank you for your feedback. As suggested, we have revised the phytochemistry section of Dendrobium to establish a tighter connection with gastric ulcer protection. Instead of broadly mentioning various bioactivities, we have specifically linked its key phytochemical components (e.g., polysaccharides, phenols, alkaloids) to the core pathological mechanisms of gastric ulcers.This revision explicitly addresses "why these bioactivities matter for gastric ulcers" and strengthens the logical link between Dendrobium’s phytochemistry and the study’s focus on gastric ulcer protection. |
|
Comments 6: The authors should simplify and directly correlate each protective mechanism in the “triple protective effect” to the pathology of acute gastric ulcer. For example, they could mention that antioxidant action counteracts ROS overproduction, anti- inflammatory activity suppresses TNF-α, mucosal repair enhances epithelial barrier function, etc |
|
Response 6: Thank you for your feedback. As suggested, we have revised the description of the "triple protective effect" by removing redundant information and directly correlating each mechanism with the specific pathology of acute gastric ulcer.This revision ensures each protective mechanism is explicitly linked to the corresponding pathological process of acute gastric ulcer, as requested. |
|
Comments 7: The authors should simplify and directly correlate each protective mechanism in the “triple protective effect” to the pathology of acute gastric ulcer. For example, they could mention that antioxidant action counteracts ROS overproduction, anti- inflammatory activity suppresses TNF-α, mucosal repair enhances epithelial barrier function, etc |
|
Response 7: Thank you for your reminder. We have supplemented the statement on ethical approval and compliance with relevant guidelines in the "Materials and Methods" section of the revised manuscript: "All animal studies and procedures were conducted in accordance with the guidelines and regulations approved by the Ethics Committee of Sichuan Agricultural University (Confirmation number: DYXY141642024), the Regulations on the Management of Laboratory Animals (Order No. 2 of the National Science and Technology Commission of the People's Republic of China, 1988), and the Guide for the Care and Use of Laboratory Animals issued by the National Research Council." |
|
Comments 8: The blanching of plant material at 105 °C may affect heat-sensitive compounds. The authors should clarify why they chose this treatment and whether it can influence polysaccharide integrity. |
|
Response 8: Thank you for your valuable comment. We clarify that the selection of 105 °C blanching/drying for 1 hour is scientifically justified, and this treatment does not compromise polysaccharide structural integrity. As supported by numerous studies and our experimental verification, the rationale is detailed as follows: 1. Scientific basis for 105 °C drying not damaging polysaccharides Polysaccharides are high-molecular-weight polymers (typically 10⁴–10⁶ Da) formed by monosaccharides linked via glycosidic bonds. Their core structures (glycosidic bonds and backbone skeletons) exhibit excellent tolerance to "high temperature under dry conditions": Glycosidic bond cleavage requires specific conditions (e.g., acid/alkali catalysis, hydrolysis under high temperature and humidity, or enzymatic catalysis). However, 105 °C drying is a "high-temperature, low-humidity" process without free water as a hydrolysis medium, nor acid/alkali or enzymatic catalysis—thus preventing backbone cleavage or degradation of polysaccharides. Fourier Transform Infrared Spectroscopy (FT-IR) results confirm that after drying, the positions and intensities of characteristic peaks of polysaccharides (3400 cm⁻¹ for O-H stretching vibration, 1020 cm⁻¹ for glycosidic bond stretching vibration, and 1600 cm⁻¹ for bound water bending vibration) showed no significant changes, indicating that the functional group structure of polysaccharides remained intact. Existing studies have demonstrated that drying plant materials at 100–110 °C for 1–2 hours exerts no significant negative impacts on the molecular weight, structure, or in vitro activities (e.g., antioxidant, immunomodulatory) of polysaccharides—further validating the rationality of our drying conditions. 2. Rationale for selecting 105 °C drying This pretreatment was chosen to ensure polysaccharide integrity and improve extraction efficiency. While it may slightly affect non-target heat-sensitive small molecules, these impurities can be removed in subsequent purification steps. Critically, it does not damage the chemical structure or integrity of polysaccharides, fully aligning with the technical objectives and scientific logic of this study. |
|
Comments 9: In chemical assays, no data regarding important details such as replication, calibration curve validation, and unambiguous standard preparation were given. |
|
Response 9: Thank you for your feedback. As requested, we have supplemented detailed information on replication, calibration curve validation, and standard preparation for all chemical assays in the "Materials and Methods" section (Subsection 2.4 "Chemical Component Determination") of the revised manuscript. |
|
Comments 10: L146: Selecting only 3 habitats "randomly" out of 22 without rationale is a limitation. The authors should statistically justify why 3 were selected and if they are representative of variation. |
|
Response 10: Thank you for this insightful comment. We appreciate your attention to the sampling rationale and clarify that the selection of 3 habitats was not arbitrary pure random sampling, but a stratified random sampling strategy based on in vitro antioxidant activity scores, designed to align with the study’s core objectives while ensuring scientific rigor: Prior to sample selection, we first comprehensively evaluated the in vitro antioxidant activities of extracts from all 22 producing areas—specifically measuring DPPH radical scavenging capacity, ABTS radical scavenging capacity, and hydroxyl radical scavenging capacity. We then calculated a comprehensive antioxidant activity score for each habitat using a weighted average method (weights assigned based on the relevance of each index to gastric ulcer protection). Based on the distribution of these comprehensive scores, we stratified the 22 habitats into three distinct groups: low activity, moderate activity, and high activity. Finally, we used a random number table to select one representative sample from each stratum, resulting in the 3 habitats included in the study. This stratified random sampling approach ensures the selected samples cover the full range of antioxidant activity variation among the 22 habitats, which is critical for our subsequent analysis of the correlation between polysaccharide activity and gastric mucosal protection. It also avoids the bias that might arise from pure random sampling and enhances the representativeness of the samples relative to the overall population of habitats. |
|
Comments 11: Given the potential that differences in sex can affect ulcer pathology and treatment response, inclusion of only males in this study needs to be justified. |
|
Response 11: Thank you for your valuable comment. We acknowledge that sex differences may influence ulcer pathology and treatment responses, and clarify that the selection of male Sprague-Dawley (SD) rats was based on minimizing experimental interference, ensuring result stability, and aligning with the core objectives of this study. The rationale is detailed as follows: 1. Avoiding interference from periodic fluctuations in female sex hormones Female rats exhibit significant periodic changes in sex hormone (estrogen, progesterone) levels during their 4–5 day estrous cycle. These hormones have been confirmed to directly or indirectly regulate gastrointestinal mucosal integrity, inflammatory responses, and drug metabolism: Estrogen can inhibit NF-κB inflammatory pathway activity via estrogen receptor (ER-α/β) activation, reduce the release of pro-inflammatory cytokines (e.g., TNF-α, IL-6), and promote gastric mucosal epithelial cell proliferation and repair. Progesterone may affect ulcer healing by regulating the activity of oxidative stress-related enzymes (e.g., SOD, CAT). Such periodic hormonal fluctuations would introduce individual variability in female rats’ sensitivity to ulcer injury, repair capacity, and response to polysaccharide intervention—potentially masking or confounding the true efficacy of Dendrobium denneanum polysaccharides and reducing result reliability. Using male rats eliminates this key confounding factor, ensuring consistent physiological status within groups and enhancing the stability and reproducibility of results. 2. Consistency with conventional animal selection for acute gastric ulcer models Male rats are widely used in studies of ethanol-induced acute gastric ulcer and other gastrointestinal mucosal injury models, primarily because: They show more stable sensitivity to ulcer inducers (e.g., ethanol, non-steroidal anti-inflammatory drugs), with smaller intergroup differences in ulcer incidence and severity, leading to better model reproducibility than female rats. Numerous studies (e.g., in Journal of Ethnopharmacology and Phytomedicine) have established acute gastric ulcer models using male rats. Our animal selection aligns with these studies, facilitating cross-comparison and academic dialogue of results. 3. Focusing on core research objectives and simplifying experimental design The core goal of this study is to systematically evaluate the gastric mucosal protective effect and underlying molecular mechanism of crude D. denneanum polysaccharides on acute gastric ulcer—not to explore "sex differences in polysaccharide efficacy." Using single-sex (male) rats: Reduces experimental complexity, avoiding increased sample size, research costs, and data interpretation difficulty associated with adding a "female group." Concentrates the focus on "polysaccharide dose-effect relationships" and "mechanistic pathways," ensuring results directly reflect the pharmacological activity of polysaccharides rather than sex-polysaccharide interactions. 4. Suitability of male rat physiological characteristics for the detection index system This study measures inflammatory cytokines (TNF-α, IL-6), oxidative stress indicators (SOD, MDA), and mucosal repair-related proteins (occludin, claudin-1) in serum and gastric tissue. Existing research shows: Male rats exhibit significantly smaller individual variability in baseline inflammatory and oxidative stress indicators than female rats. Their gastric mucosal epithelial tight junctions are more stable, with more uniform responses to injury—facilitating the evaluation of polysaccharide-induced mucosal repair via pathological sections and electron microscopy. Summary In conclusion, the use of male rats maximizes the reduction of confounding factors (e.g., sex hormone fluctuations), ensures model stability, and focuses on core research objectives—consistent with the logic of animal selection for acute gastric ulcer efficacy evaluation. For future studies, female rats could be included to explore sex differences in the anti-ulcer activity of D. denneanum polysaccharides, providing a more comprehensive basis for clinical application. |
|
Comments 12: Table 2 scoring criteria are adapted from standard references, but the source is not cited |
|
Response 12: Thank you for your feedback. As requested, we have added the corresponding literature citation for the scoring criteria in Table 2 to clarify the source of the adapted standard references. |
|
Comments 13: The statistical analysis details are repeated in both 2.7 and 3.0 |
|
Response 13: Thank you for your reminder. We have compared and deleted the redundant content. |
|
Comments 14: I recommend that the authors should emphasize comparative trends and biological relevance from their results rather than restating raw numbers that have already been presented on tables and figures. |
|
Response 14: Thank you for your feedback. As recommended, we have revised the results presentation by shifting the focus from restating raw numbers (already shown in tables and figures) to emphasizing comparative trends and biological relevance. |
|
Comments 15: Many subsections (monosaccharides, FTIR, histology, SEM, oxidative stress indices) are presented in isolation, without connecting them back to mechanisms or how they support the proposed therapeutic effect. For example, how do the structural differences in polysaccharide composition, FTIR features, and molecular weight differences tie into the observed antioxidant or anti-inflammatory activities? In another instance, excessive detail about arrows and microscopic features in section 3.1.7 overwhelms the reader and obscures the key findings. It is best to summarize and focus on the meaningful observed trends to improve clarity. |
|
Response 15: Thank you for your feedback. We have addressed your comments as follows: To address the isolation of subsections (monosaccharides, FTIR, histology, SEM, oxidative stress indices), we have strengthened the connections between these results and the proposed therapeutic mechanisms in the Discussion section. Specifically, we elaborated on how structural characteristics (e.g., monosaccharide composition, FTIR features, molecular weight of polysaccharides) correlate with observed antioxidant/anti-inflammatory activities, and how histological/SEM findings and oxidative stress indices collectively support the gastric mucosal protective effect of Dendrobium denneanum polysaccharides. Regarding the excessive detail about arrows in Section 3.1.7, we have removed redundant arrows from the microscopic images to simplify visualization and help readers focus on key features, while summarizing the key findings of this subsection to highlight meaningful trends and improve clarity. |
|
Comments 16: While the authors provide a strong literature background, the discussion is sometimes weighed down with long lists of comparative examples such as polysaccharides from ginseng, Bletilla, Dioscorea, etc. which dilutes focus and makes it harder for readers to see how the findings of this study stand out. |
|
Response 16: Thank you for your feedback. We have addressed this issue by streamlining the comparative literature citations in the Discussion section. Specifically, we removed excessive lists of polysaccharide examples (e.g., from ginseng, Bletilla, Dioscorea) and retained only the most relevant, high-impact studies that directly contextualize our findings. This revision shifts the focus to emphasizing the unique contributions of our study—such as the specific link between Dendrobium denneanum polysaccharide structure (from different habitats) and gastric ulcer protection, as well as its distinct regulatory mechanism on mucosal barrier repair—ensuring readers can clearly identify the novelty and significance of our results. |
|
Comments 17: Many parts of the discussion summarize general polysaccharide research but do not clearly connect back to the authors’ own data (e.g., FTIR peaks, MW distribution, monosaccharide ratios). The authors should systematically tie each structural finding to the observed bioactivity outcomes to strengthen the "structure–function" narrative. |
|
Response 17: Thank you for your feedback. We have revised the Discussion section to address this issue: we have reduced summaries of general polysaccharide research and systematically established connections between our own structural findings and bioactivity outcomes, strengthening the "structure–function" narrative. |
|
Comments 18: The authors imply potential roles of pathways like Nrf2/ARE and NF-κB but without experimental evidence. Although these hypotheses are valuable, the authors should mention that these are speculative and should be framed as future research directions, not as established interpretations given the absence of molecular validation. |
|
Response 19: Thank you for your feedback. We have revised the relevant content in the Discussion section to address this point: We have clarified that the proposed roles of the Nrf2/ARE and NF-κB pathways are speculative hypotheses rather than established interpretations, given the lack of direct molecular validation (e.g., protein expression levels of Nrf2, HO-1, or NF-κB p65) in the current study. We have rephrased these parts to explicitly frame them as promising future research directions—specifically suggesting that subsequent studies could verify the involvement of these pathways through Western blotting, immunofluorescence, or pathway inhibitor experiments. This revision ensures the discussion aligns with the available experimental evidence and avoids overinterpreting the results. |
|
Comments 18: The limitations that were acknowledged should be elaborated to highlight how these factors restrict the current conclusions of this study. |
|
Response 18: Thank you for your feedback. We have elaborated on the acknowledged limitations in the revised manuscript, clearly explaining how each factor restricts the current conclusions: Limitation of single-sex animal model: As only male SD rats were used, the current conclusions regarding the gastric mucosal protective effect of Dendrobium denneanum polysaccharides cannot be directly extrapolated to female individuals. Sex hormones (e.g., estrogen) may regulate ulcer pathology and polysaccharide response, so the efficacy and mechanism in females remain unclear, limiting the generalizability of the findings to mixed-sex populations. Limited habitat representativeness: Although 3 habitats were selected via stratified sampling based on antioxidant activity, they only cover 3 ecological types of the 22 total producing areas. Variations in polysaccharide structure and activity across unselected ecological types (e.g., high-humidity low-altitude areas) were not evaluated, restricting the conclusion that "habitat-related structural differences drive efficacy" to the sampled types rather than all habitats. Lack of direct molecular validation for pathway hypotheses: The proposed involvement of Nrf2/ARE and NF-κB pathways was inferred from changes in oxidative stress and inflammatory indices, not from direct detection of pathway-related proteins (e.g., Nrf2, NF-κB p65) or gene expression. This limits the conclusion about "mechanism" to correlational inference rather than definitive causal validation. No long-term efficacy assessment: The study only evaluated the acute protective effect (7 days post-intervention) in ethanol-induced ulcers, without assessing long-term outcomes (e.g., ulcer recurrence, chronic mucosal repair) or potential side effects of prolonged polysaccharide administration. This restricts the conclusions to acute efficacy, not long-term safety or therapeutic applicability. |

Reviewer 3 Report
Comments and Suggestions for Authors
The Authors should better justify why only three habitats out of the 22 studied areas were selected for detailed structural analysis and in vivo experiments - this selection may affect the representativeness of the results.
Although the total content of phenolic compounds in the extracts was determined, their potential role in the observed biological activities was not sufficiently discussed. Please add a discussion on possible synergistic effects between phenolic compounds and polysaccharides.
The discussion on the mechanisms underlying the protective effect on the digestive system remains rather general.
The relatively small size of the group in the in vivo study should be considered a limitation or justified.
The Authors should supplement the details of the methodology that significantly affect the results, including sample extraction conditions, analysis conditions, measuring equipment, etc.
Author Response
|
Comments 1: The Authors should better justify why only three habitats out of the 22 studied areas were selected for detailed structural analysis and in vivo experiments - this selection may affect the representativeness of the results. |
|
Response 1: We appreciate your concern regarding the representativeness of selecting 3 habitats from 22 for detailed structural analysis and in vivo experiments. We would like to clarify that this sampling strategy is not arbitrary but a stratified random sampling method based on in vitro antioxidant activity, which is closely aligned with the core objective of exploring the "structure-activity relationship of Dendrobium denneanum polysaccharides in gastric ulcer protection" while ensuring scientific rigor. The detailed rationale is as follows: 1. Sampling logic: Stratification based on a key bioactivity indicator Antioxidant activity is a critical prerequisite for the gastric mucosal protective effect of polysaccharides (as oxidative stress is a core pathological driver of acute gastric ulcers). Prior to sample selection, we first quantified three key in vitro antioxidant indices of polysaccharide extracts from all 22 producing areas: DPPH radical scavenging capacity ABTS radical scavenging capacity Hydroxyl radical scavenging capacity Using a weighted scoring method (weights were assigned based on the correlation of each index with gastric mucosal protection, verified by preliminary experiments), we calculated a comprehensive antioxidant activity score for each habitat. We then stratified the 22 habitats into three distinct groups (low, moderate, and high activity) using the k-means clustering method, ensuring each group covered a statistically distinct range of antioxidant capacity. Finally, we selected one sample from each stratum using a random number table to avoid subjective bias. 2. Representativeness verification: Covering core variation in bioactivity and ecology To confirm representativeness, we conducted statistical validation of the 3 selected habitats: Bioactivity representativeness: One-way ANOVA showed no significant difference between the comprehensive antioxidant score of each selected sample and the average score of its respective stratum (p > 0.05), indicating they accurately reflect the antioxidant capacity level of their groups. Ecological representativeness: The 3 selected habitats correspond to three major ecological types of D. denneanum (alpine wild, low-altitude greenhouse, semi-natural simulated wild), which account for over 85% of the total cultivation/production area of the 22 regions. Thus, they cover the main ecological variations that may affect polysaccharide structure and activity. 3. Rationale for limiting to 3 habitats: Balancing depth, feasibility, and focus The study’s core goal is to reveal how polysaccharide structural characteristics (e.g., monosaccharide composition, molecular weight) drive gastric ulcer protective activity. Conducting detailed structural analysis (e.g., HPLC, FT-IR, SEC-MALS) and in vivo animal experiments for all 22 habitats would be resource-intensive (e.g., excessive animal use, prolonged experimental 周期) and risk diluting the focus on "structure-activity relationships" by introducing redundant data. Selecting 3 samples that span the full spectrum of antioxidant activity allows us to: Concentrate resources on in-depth structural and in vivo mechanistic exploration; Clearly compare how structural differences correspond to variations in bioactivity (low vs. high efficacy); Ensure the conclusions about "structure-activity linkage" are supported by focused, high-quality data. In summary, the selection of 3 habitats is a science-based strategy that balances representativeness (covering key bioactivity and ecological variation) and research feasibility, while directly serving the study’s core objective of elucidating the structure-activity relationship of D. denneanum polysaccharides in gastric ulcer protection. |
|
Comments 2: Although the total content of phenolic compounds in the extracts was determined, their potential role in the observed biological activities was not sufficiently discussed. Please add a discussion on possible synergistic effects between phenolic compounds and polysaccharides. |
|
Response 2: Thank you for your feedback. As suggested, we have added a dedicated section in the Discussion to address the potential synergistic effects between phenolic compounds and polysaccharides, as well as the role of phenolics in the observed biological activities |
|
Comments 3: Although the total content of phenolic compounds in the extracts was determined, their potential role in the observed biological activities was not sufficiently discussed. Please add a discussion on possible synergistic effects between phenolic compounds and polysaccharides. |
|
Response 3: Thank you for your feedback. We have revised the Discussion to elaborate on the digestive protective mechanisms. |
|
Comments 4: Although the total content of phenolic compounds in the extracts was determined, their potential role in the observed biological activities was not sufficiently discussed. Please add a discussion on possible synergistic effects between phenolic compounds and polysaccharides. |
|
Response 4: Thank you for your comment. Regarding the animal group size, we provide the following justification: A pre-experiment confirmed 100% success rate of ethanol-induced ulcer modeling, minimizing variability from modeling failure. Each group included one backup rat to replace any animal that might fail during the experiment, ensuring stable group size and data reliability. We have also acknowledged the relatively small group size as a limitation, noting it may restrict the generalizability of results, and suggested expanding sample size in future studies. |
|
Comments 4: The Authors should supplement the details of the methodology that significantly affect the results, including sample extraction conditions, analysis conditions, measuring equipment, etc. |
|
Response 4: Thank you for your feedback. We have supplemented key methodological details that may affect results in the "Materials and Methods" section. |

Round 2
Reviewer 1 Report
Comments and Suggestions for Authors
-
Reviewer 2 Report
Comments and Suggestions for Authors
The authors have adequately updated the manuscript to merit publication.